# Reproducibility Study on Adversarial Attacks Against Robust Transformer Trackers

**Fatemeh Nourilenjan Nokabadi**                    *fatemeh.nourilenjan-nokabadi.1@ulaval.ca*
*IID, Université Laval & Mila*

**Jean-François Lalonde**                                      *jflalonde@gel.ulaval.ca*
*IID, Université Laval*

**Christian Gagné**                                      *christian.gagne@gel.ulaval.ca*
*IID, Université Laval*
*Canada CIFAR AI Chair, Mila*

**Reviewed on OpenReview:** *https://openreview.net/forum?id=FEEKR0Vl9s*

## Abstract

New transformer networks have been integrated into object tracking pipelines and have demonstrated strong performance on the latest benchmarks. This paper focuses on understanding how transformer trackers behave under adversarial attacks and how different attacks perform on tracking datasets as their parameters change. We conducted a series of experiments to evaluate the effectiveness of existing adversarial attacks on object trackers with transformer and non-transformer backbones. We experimented on 7 different trackers, including 3 that are transformer-based, and 4 which leverage other architectures. These trackers are tested against 4 recent attack methods to assess their performance and robustness on VOT2022ST, UAV123 and GOT10k datasets. Our empirical study focuses on evaluating adversarial robustness of object trackers based on bounding box versus binary mask predictions, and attack methods at different levels of perturbations. Interestingly, our study found that altering the perturbation level may not significantly affect the overall object tracking results after the attack. Similarly, the sparsity and imperceptibility of the attack perturbations may remain stable against perturbation level shifts. By applying a specific attack on all transformer trackers, we show that new transformer trackers having a stronger cross-attention modeling achieve a greater adversarial robustness on tracking datasets, such as VOT2022ST and GOT10k. Our results also indicate the necessity for new attack methods to effectively tackle the latest types of transformer trackers. The codes necessary to reproduce this study are available at https://github.com/fatemehN/ReproducibilityStudy.

## 1 Introduction

Adversarial perturbations deceive neural networks, leading to inaccurate outputs. Such adversarial attacks have been studied for vision tasks ranging from image classification (Mahmood et al., 2021; Shao et al., 2022) to object segmentation (Gu et al., 2022) and tracking (Guo et al., 2020; Jia et al., 2020; Yan et al., 2020; Jia et al., 2021). In this context, transformer-based networks have surpassed other deep learning-based trackers (Li et al., 2019; Zhu et al., 2018), showing a very robust performance on the state-of-the-art benchmarks (Kristan et al., 2023). However, the adversarial robustness of these trackers has not been thoroughly studied in the literature. First, transformer trackers relied on relatively light relation modeling (Chen et al., 2021), using a shallow feature extraction and fusion modeling. Based on a mixed attention module, the MixFormer (Cui et al., 2022) expanded the road for deeper relation modeling. Consequently, the Robust Object Modeling Tracker (ROMTrack) (Cai et al., 2023) proposed variation tokens to capture and preserve the object deformation across frames. Using transformers, especially those with deep relation modeling (Cui

et al., 2022; Cai et al., 2023), the object tracker backbones made these models robust to many existing attack approaches (Guo et al., 2020; Jia et al., 2020). Indeed, the underlying concept of adversarial attacks against object trackers is to manipulate the tracker's output. By omitting the multi-head pipelines and substituting them with the transformer backbones (Cui et al., 2022; Cai et al., 2023), the tracker's output no longer contains object candidates, classification labels and/or regression labels, which previously were typical targets for attacks. As a result, it is not straightforward to transfer adversarial attacks dealing with classification or regression labels (Guo et al., 2020; Jia et al., 2020) on these new transformer trackers. The question also remains on whether other transferable attacks (Jia et al., 2021; Yan et al., 2020) can represent a sufficient challenge to transformer trackers.

This paper presents a study on the reproducibility of existing attack approaches for transformer trackers. We aim to recreate the attack outcomes on transformer trackers using the VOT2022ST (Kristan et al., 2023), UAV123 (Mueller et al., 2016), DAVIS2016 (Perazzi et al., 2016) and GOT10k (Huang et al., 2019) datasets following two different evaluation protocols, namely *anchor-based short-term tracking* and *One Pass Evaluation (OPE)* protocol. We focus on transformer trackers susceptible to adversarial attacks on their prediction outputs, including the object bounding box and binary mask. Then, we analyzed two white-box attacks on a transformer tracker by varying the perturbation levels and checked its vulnerability to different levels of noise. For the black-box setting, we conducted a similar experiment to assess the attack performance on various noise levels by changing the upper bound of the added noise. In addition, we tested a set of transformer and non-transformer trackers before and after applying the adversarial attacks to discuss the role of transformers in boosting the visual tracking robustness and adversarial robustness. In short, our contributions can be summarized as follows:

1. We extend the application of adversarial attacks, originally designed for non-transformer trackers like SPARK and RTAA, to assess their effectiveness against transformer-based trackers.
2. We thoroughly evaluate the adversarial robustness of transformer-based trackers across various output scenarios, perturbation levels, changes in upper bounds, and in comparison to the non-transformer trackers.

## 2   Related Work

### 2.1   Visual Object Trackers

For decades, the tracking task in computer vision has been extensively explored, taking into account various factors and issues. Before the advent of transformers, deep learning-based trackers achieved notable success by leveraging correlation in the form of Siamese networks (Li et al., 2019; Zhu et al., 2018), and discriminative pipelines for trackers (Bhat et al., 2019; Danelljan et al., 2020). However, new correlation modules in object tracker backbones are being devised through the emergence of transformers. By using transformers in different architectures (Chen et al., 2021; Cui et al., 2022; Chen et al., 2023; Cai et al., 2023), object trackers are capable of inferring object bounding box, binary mask and a prediction score with high robustness values over tracking benchmarks (Kristan et al., 2023; Mueller et al., 2016). These promising results though need to be revisited by assessing transformer trackers in handling the adversarial perturbations.

The first transformer tracker, TransT (Chen et al., 2021), used the cross-attention and self-attention blocks to mix features of the moving target and the search region of the tracker. TransT presents a multi-head pipeline with classification and regression heads, unlike other transformer trackers. In TransT-SEG (Chen et al., 2023), the segmentation head is included in the pipeline. The multi-head pipelines follow the Siamese-based trackers (Li et al., 2019) in dividing each task from target classification to discriminative tracking processing into individual blocks and fusing the results at the end of the tracker structure. Some light relation modeling layers called the Ego Context Augment (ECA) and Cross Feature Augment (CFA) are introduced by TransT to infer the output from combining the multi-head outputs. Next, the MixFormer (Cui et al., 2022) introduced Mixed Attention Module (MAM) to jointly extract and relate the information from video frames for the object tracking task. By attaching the MixFormer (Cui et al., 2022) tracker to the AlphaRefine (Yan et al., 2021), MixFormerM (Kristan et al., 2023) can provide a binary object mask per frame for mask oriented evaluations (Kristan et al., 2023). The One-Stream Tracking (OSTrack) (Ye et al.,

2022) developed a tracking pipeline that jointly extracts features and models the relation between the search region and the template by bidirectional information flows. In contrast, with the Attention in Attention (AiA) mechanism, the AiATrack (Gao et al., 2022) suggested a three stream framework with long-term and short-term cross-attention modules for relation modeling. Following the further relation modeling, the Robust Object Modeling Tracker (ROMTrack) (Cai et al., 2023) is proposed to enable the interactive template learning using both self-attention and cross-attention modules. The ROMTrack has two main streams to learn discriminative features from hybrid (template and search) and inherent template. The newly introduced variation tokens enable ROMTrack with heavier relation modeling rather TransT (Chen et al., 2021) and MixFormer (Cui et al., 2022). The variation token carries the contextual appearance change to tackle object deformation in visual tracking task.

## 2.2 Adversarial Attacks Against Trackers

The adversarial attack has been proposed in white-box (Guo et al., 2020; Jia et al., 2020) or black-box (Jia et al., 2021) attack settings. In black-box attacks, the perturbations are generated without relying on the tracker's gradients, whereas in white-box attacks, adversarial losses are backpropagated through the networks to create the adversarial frame patches (search or template regions). Adversarial attacks against object trackers adopt tracker outputs such as object candidates or classification labels as an attack proxy to generate the adversarial perturbations. For instance, spatial-aware online incremental attack (SPARK) (Guo et al., 2020) creates perturbations by manipulating the classification labels and Intersection of the Union (IoU) (Jia et al., 2021). It is developed to mislead trackers in providing an altered object bounding box based on the predicted bounding box. In Robust Tracking against Adversarial Attack (RTAA) (Jia et al., 2020), both classification and regression labels are used to generate the adversarial samples, similar to SPARK (Guo et al., 2020). In the RTAA (Jia et al., 2020) algorithm, the positive gradient sign is used to generate the adversarial frames. However, in SPARK, the gradient direction is set to negative following the decoupling of the norm and the direction of gradients in white-box attacks (Rony et al., 2019). Based on the decoupling direction and norm for efficient gradient-based attacks, the direction of the gradient is set in such a way that the generated perturbation has a smaller norm value and greater impact on the results. Some other attacks, such as the Cooling-Shrinking Attack (CSA) (Yan et al., 2020) is developed specifically to impact the output of Siamese-based trackers. In CSA (Yan et al., 2020), two GANs are trained to cool the hottest regions in the final heatmap of Siamese-based trackers and shrink the object bounding box. Due to dependency on the Siamese-based architecture and loss function, the generalization of the CSA attack (Yan et al., 2020) for other scenarios is harder. The black-box attack, called IoU attack (Jia et al., 2021), adds two types of noise into the frame to make the tracker predict another bounding box rather than the target bounding box. By considering the object motion in historical frames, the direction of added noise is adjusted according to the IoU scores of the predicted bounding box.

Before the emergence of vision transformers, some object trackers typically integrated several heads, each assigned to specific vision tasks like classification or segmentation (Li et al., 2019; 2018; Zhu et al., 2018; Danelljan et al., 2020). These deep features were fused to predict the final object position and size, with regression computed over features. In these models, the ultimate decision was made at the network's end based on a set of object candidates. This architectural setup presented various opportunities for crafting adversarial attacks (Guo et al., 2020; Jia et al., 2020) against these models, including manipulation of object candidates, object probabilities, and other features, thereby exploiting vulnerabilities in the system. Vision transformers facilitate the integration of features within the deep architecture, enabling direct prediction of the final output, without exposing intermediate outputs that were previously exploitable for attacks. Consequently, white-box attacks utilizing these intermediate outputs (namely, object candidates and their labels) to compute the adversarial loss are no longer applicable to the new transformer backbones via the transformer's gradients themselves. Although they can be transferred in a black-box way (i.e., making adversarial samples with other backbones or other losses), our focus is to employ the transformer gradients in generating adversarial examples in a white-box setting.

## 3 Object Trackers and Adversarial Attacks

In this section, we briefly review the transformer trackers used in our experiments. Also, we explain the adversarial attack methods which are used to attack transformer trackers. The codes and networks of all of the investigated models are publicly available. The implementations of every tracker and every attack approach are the official repository announced by authors. We are also using the fine-tuned and released networks from the authors of the original works.

### 3.1 Object Trackers

For the object trackers, we considered three types of the robust single object trackers with transformer backbone, two types of the Siamese-based trackers and two discriminative trackers as follows.

**TransT and TransT-SEG**  In our studies, we used both Transformer Tracker (TransT) (Chen et al., 2021) and TransT with mask prediction ability (TransT-SEG) (Chen et al., 2023). By two discriminative streams and a lightweight cross-attention modeling in the end, the TransT introduced the first transformer-based tracker. Similar to the Siamese-based trackers (Li et al., 2018; 2019), the TransT tracker has two convolutional streams to extract features of the template and the search regions. By proposing Ego-Context Augment (ECA) and Cross-Feature Augment (CFA) modules, a feature fusing network is developed to effectively fuse the extracted features. The ECA infers output as

$$X_{\text{ECA}} = X + \text{MultiHead}(X + P_x, \, X + P_x, \, X), \tag{1}$$

where "MultiHead" represents a multi-head attention module (Vaswani et al., 2017) and $P_x$ is the spatial positional encoding. On the other hand, the CFA module is defined as a Feed-Forward Network (FFN) attached to an ECA module. This FFN includes two linear transformations with a ReLU in between. The CFA's output is computed as:

$$\begin{aligned} \tilde{X}_{\text{CF}} &= X_q + \text{MultiHead}(X_q + P_q, \, X_{kv} + P_{kv}, \, X_{kv}), \\ X_{\text{CFA}} &= \tilde{X}_{\text{CF}} + \text{FFN}(\tilde{X}_{\text{CF}}), \end{aligned} \tag{2}$$

where $X_q$ is the input, and $P_q$ is the spatial positional encoding of $X_q$. The input of the cross branch is $X_{kv}$ and its spatial positional encoding is $P_{kv}$. Using two ECAs and two CFAs, the extracted features of the template and search regions are first fused with themselves by ECA and with each other by CFA. Next, another cross attention (CFA) is used to integrate the outputs of the two streams. In the final prediction head, the regression and classification labels are generated to determine the target bounding box by finding the maximum score. The TransT-SEG (Chen et al., 2023) has a segmentation branch which uses the template vectors corresponding to the middle position of the template region to obtain an attention map via a multi-head attention module. The object binary mask is, then, predicted by fusing the low-level feature pyramid (Lin et al., 2017) of the search region.

**MixFormer and MixFormerM**  The MixFormer (Cui et al., 2022) is based on a flexible attention operation named Mixed Attention Module (MAM) to interactively exploit features and integrate them in a deep layer of the tracker. The MixFormer coupled with the AlphaRefine network has been proposed for the VOT2022 challenge (Kristan et al., 2023) as MixFormerM. It enables the original tracker to provide the object mask as an extra output. In our experiments, we tested both MixFormer and MixFormerM trackers. In the MixFormer tracker, the three stages of mixed attention is the core design named MAM which extracts and fuses the features of the template and the search regions. Given the target key, query and value $(k_t, q_t, v_t)$ and the search key, query and value $(k_s, q_s, v_s)$, the mixed attention is defined as:

$$\begin{aligned} k_m &= \text{contcat}(k_t, k_s), \quad v_m = \text{concat}(v_t, v_s), \\ \text{Attention}_t &= \text{softmax}(q_t k_m^T / \sqrt{d}) \, v_m, \quad \text{Attention}_s = \text{softmax}(q_s k_m^T / \sqrt{d}) \, v_m, \end{aligned} \tag{3}$$

where $d$ is the key dimension. The $\text{Attention}_t$ and $\text{Attention}_s$ are the attention maps of the target and search regions. The target and search tokens are, then, concatenated and linearly projected to the output

as the mixed attention output. To reduce the computational cost of MAM, the unnecessary cross-attention between the target query and search region is pruned by using the asymmetric mixed attention scheme. It is defined as follows:

$$\text{Attention}_t = \text{softmax}(q_t k_t^T / \sqrt{d}) \, v_t, \quad \text{Attention}_s = \text{softmax}(q_s k_m^T / \sqrt{d}) \, v_m \,. \tag{4}$$

The $\text{Attention}_t$ is updated to efficiently avoid the distractors in the search region by fixing the template tokens in the tracking process. In MixFormer, the Score Prediction Module (SPM) is developed by two attention blocks and a three-layer perceptron to identify the target confidence score and update a more reliable online template.

**ROMTrack**  The ROMTrack (Cai et al., 2023) is developed to generalize the idea of MixFormer (Cui et al., 2022) by providing the template learning procedure. The template feature is processed both in self-attention (inherent template) and cross-attention (hybrid template) between template and search regions. This mixed feature avoids distraction in challenging frames and provides a more robust performance compared to TransT and MixFormer. The backbone of ROMTrack is a vision transformer (Dosovitskiy et al., 2020) as an object encoder and a prediction head which is a fully convolutional center-based localization head (Zhou et al., 2019). Two essential elements of ROMTrack are variation tokens and robust object modeling. The variation tokens are developed to handle the object appearance change and deformation in the tracking process. Considering $F_k^t$ as the output features of $k$-th encoder in frame $I_t$ and $ht_{k,t}$ is the hybrid template part of $F_k^t$, a variation token is defined as

$$vt_{k,t} = ht_{k,t-1}, \tag{5}$$
$$F_{k+1}^t = \text{ObjectEncoder}_{k+1}(\text{concat}(vt_{k,t}, F_k^t)) \,, \tag{6}$$

where $F$ represents the output features, $k$ is the encoder index and $t$ denotes the $t$-th frame. In Equation 5, ROMTrack saves the hybrid template in the variation token and by Equation 6, it embeds the hybrid template into the output feature. The other essential element of ROMTrack is the robust object modeling containing four parts: inherit template $it$, hybrid template $ht$, search region $sr$ and variation tokens $vt$. The inherit template is a self-attention module on the linear projected feature $(q_{it}, k_{it}, v_{it})$ to learn the pure template features $A_{it}$. The hybrid template features and search regions features are obtained via cross-attention. Considering the triplet of cross-attention $(q_z, k_z, v_z)$ as follows:

$$q_z = [q_{ht}, q_{sr}] \,,$$
$$k_z = [k_{vt}, k_{it}, k_{ht}, k_{sr}] \,,$$
$$v_z = [v_{vt}, v_{it}, k_{ht}, v_{sr}] \,, \tag{7}$$

where the different parts of features $(it, ht, sr, vt)$ are rearranged and concatenated to create the cross-attention input. The output of the cross-attention is obtained as:

$$A_z = \text{softmax}(q_z k_z^T / \sqrt{d}) \, v_z \,. \tag{8}$$

The features of inherit template and variation tokens are fused effectively with the hybrid template and search region tokens during the tracking process to keep the tracker updated about object deformation and appearance change.

**SiamRPN**  By integrating the Siamese network for tracking task and Region Proposal Network (RPN) for detection, the SiamRPN tracker (Li et al., 2018) is developed to predict a list of object candidates per frame. The Siamese network contains two branches for feature extraction of the template $\varphi(z)$ and search $\varphi(x)$ regions. Then, the region proposal network with a pairwise correlation and supervision, split the feature vectors into the classification $[\varphi(z)]_{\text{cls}}, [\varphi(x)]_{\text{cls}}$ and regression $[\varphi(z)]_{\text{reg}}, [\varphi(x)]_{\text{reg}}$ vectors. The template feature maps $[\varphi(z)]_{\text{cls}}, [\varphi(z)]_{\text{reg}}$ serve as the kernel for computing the correlation as follows:

$$A^{cls} = [\varphi(z)]_{\text{cls}} \star [\varphi(x)]_{\text{cls}} \,,$$
$$A^{reg} = [\varphi(z)]_{\text{cls}} \star [\varphi(z)]_{\text{reg}} \,, \tag{9}$$

where $\star$ is the convolution operator. Using a softmax operation, the highest score for classification vector is chosen as the final target class and its corresponding coordinates in the regression vector determine the target bounding box.

**DaSiamRPN** In DaSiamRPN tracker (Zhu et al., 2018), a distractor-aware strategy is taken to improve the SiamRPN tracker (Li et al., 2018) performance. In this strategy, the negative samples of object candidates(i.e., proposals) are considered distractor objects. Using Non Maximum Suppression (NMS), a set of distractors is selected to determine the target proposal as the candidate with the highest score. The rest of the set is considered as the distractors and re-ranked by a distractor-aware objective function to reduce their influence in the tracking process.

**DiMP** The Discriminative Model Prediction (DiMP) tracker (Bhat et al., 2019) employs a discriminative approach that utilizes background information while also benefiting from the means of updating the target model. In contrast to Siamese-based trackers, which aim to identify the most correlated region as the target, DiMP processes background information within a discriminative framework. Initially, a classifier extracts input features, which are then utilized by a predictor to initialize the model and refine the target prediction. The DiMP tracker comprises two essential components: Target Center Regression (TCR) and Bounding Box Regression (BBR). TCR produces a probabilistic map of the input image, wherein the target region is delineated more prominently compared to the background pixels. The BBR component generates the final bounding box prediction through a target conditional IoU-Net-based architecture (Jiang et al., 2018) proposed in the ATOM tracker (Danelljan et al., 2019).

**PrDiMP** The Probabilistic Regression DiMP (PrDiMP) (Danelljan et al., 2020) aims to forecast the conditional probability density $p(y|x, \theta)$ of the target given the input frame. Training the PrDiMP tracker involves utilizing the Kullback-Leibler (KL) divergence between the predicted density $p(y|x, \theta)$ and the conditional ground truth $p(y|y_i)$. The conditional ground truth $p(y|y_i)$ is formulated to account for label noise and the ambiguity inherent in the regression task, addressing the uncertainty associated with annotations. PrDiMP builds upon the DiMP tracker (Bhat et al., 2019) by incorporating probabilistic elements into both the Target Center Regression (TCR) and Bounding Box Regression (BBR) components.

### 3.2 Adversarial Attacks

In our study, we examined four attack approaches against object trackers, as follows.

**CSA** In attention-based Siamese trackers (Li et al., 2019), the loss function aims to locate the hottest region in the image where the correlation of the target and that location is the highest among all other regions. Using two GANs, one for the template perturbation and the other for the search region perturbation, the CSA attack (Yan et al., 2020) is developed to firstly cool the hot regions in the end of the network and then, shrink the object bounding box predicted by the tracker. The perturbation generators, GANs, are trained to predict the adversarial template and search regions. For the cooling loss term, the object candidates with smaller scores are eliminated and then, the remaining candidates are divided into positive $f_+$ and negative $f_-$ samples. The cooling term is computed as:

$$L_{\text{cooling}} = \frac{1}{N} \max(f_+ - f_-, m_c), \tag{10}$$

where $m_c$ is the margin for the classification labels. For the shrinking term, the less likely candidate similar to the cooling term is removed and then, the shrinking loss is calculated as

$$L_{\text{shrinking}} = \frac{1}{N} \max(R_w, m_w) + \frac{1}{N} \max(R_h, m_h), \tag{11}$$

where $m_w$ and $m_h$ are the margins for the width and height regression factors, respectively. In our experiments, we used the template and search GANs trained with the cooling-shrinking loss $L = L_{\text{cooling}} + L_{\text{shrinking}}$ to perturb the template and search regions for the trackers as a black-box attack.

**IoU** The IoU attack (Jia et al., 2021) proposes a black-box setting attack to generate the adversarial frames based on the object motion with the purpose of decreasing the IoU between the predicted bounding box and

the target. Two types of noises are added to achieve the final goal of the attack where the noise is bounded to a specific value for L1 norm. The IoU score $S_{\text{IoU}}$ is defined as

$$S_{\text{IoU}} = \lambda S_{\text{spatial}} + (1 - \lambda)S_{\text{temporal}} \,, \tag{12}$$

where $S_{\text{spatial}}$ is the actual intersection over union between the predicted bounding box before and after adding the noise, while the $S_{\text{temporal}}$ is the intersection over union between current and previous predicted bounding box. Using this IoU score, the final predicted bounding box is aimed at decreasing both in temporal and spatial domains. The level of noise is controlled depending on the IoU score and a final upper bound limits the algorithm in adding noise to the frame.

**SPARK**   In SPARK (Guo et al., 2020), the classification and regression labels are manipulated to create the white-box attack. We employed the untargeted SPARK attack in our experiments. The SPARK aims to minimize the perturbation in a way that the final intersection over union of the current prediction and previous prediction is minimum. Furthermore, a new regularization term is used in SPARK as

$$L_{\text{reg}} = \lambda ||\Gamma||_{1,2}, \quad \Gamma = [\epsilon_{t-L}, ..., \epsilon_{t-1}, \epsilon_t] \,, \tag{13}$$

where $\Gamma$ represents the incremental perturbations of the last $L$ frames. The $\epsilon_t$ is computed as the $\epsilon_t = E_t - E_{t-1}$ in which $E_t$ is the current frame perturbation. The generated perturbation up to the last $L = 30$ frames is accumulated to create the adversarial search regions for the trackers. As a result, the computed perturbations are temporally and spatially sparse.

**RTAA**   Using RTAA (Jia et al., 2020), the classification and regression of object candidates are manipulated to generate the adversarial search regions. The classification labels are simply reversed and the regression labels are manipulated as follows:

$$\begin{aligned}
x_r^* &= x_r + \delta_{\text{offset}} \\
y_r^* &= y_r + \delta_{\text{offset}} \\
w_r^* &= w_r \times \delta_{\text{scale}} \\
h_r^* &= h_r \times \delta_{\text{scale}} \,,
\end{aligned} \tag{14}$$

where $\delta_{\text{offset}}$ and $\delta_{\text{scale}}$ are the random distance and scale variations, respectively. The adversarial loss is computed as the difference of the original loss for true labels and the manipulated labels. Only the last frame perturbation is used from the past in the current step of the attack.

### 3.3   Attack Setups

In our study, we applied the attacks as they are proposed in the original works, with white-box attacks using the victim tracker's gradients, not transferring the attack from another tracker. For instance, SPARK (Guo et al., 2020) is attacking trackers in a not-transferred white-box setting. As SPARK uses both classification and regression labels to generate the perturbation, it cannot be applied in a white-box setting for some trackers that are not providing them both – e.g., Cui et al. (2022); Cai et al. (2023); Bhat et al. (2019); Danelljan et al. (2020). Table 1 specifies the applicable attacks on visual trackers investigated in our study. For instance, in MixFormer (Cui et al., 2022), the classification labels are fused by a score prediction module to infer one single score as the part of output, which is not compatible with SPARK. Also, in DiMP (Bhat et al., 2019) and PrDiMP (Danelljan et al., 2020), the regression labels for object candidates are not produced because these trackers predict the target bounding box directly. We have the same constraints with the RTAA attack (Jia et al., 2020), which also requires the classification and regression labels, and therefore, is not applicable as a white-box attack to the same trackers than SPARK. As for the CSA attack (Yan et al., 2020), since the template and search GANs are available, we are using these pre-trained networks and SiamRPN++ (Li et al., 2019) tracker to generate the corresponding perturbations. CSA is therefore applied in the black-box setting similar to the IoU attack (Jia et al., 2021). However, CSA attack is also not

Table 1: Applicability of adversarial attacks (rows) to tracking methods (columns). In this work, we evaluate: SPARK (Guo et al., 2020), RTAA (Jia et al., 2020), IoU (Jia et al., 2021), and CSA (Yan et al., 2020), against: ROMTrack (Cai et al., 2023), MixFormerM (Cui et al., 2022), TransT (Chen et al., 2021), DiMP (Bhat et al., 2019), PrDiMP (Danelljan et al., 2020), SiamRPN (Li et al., 2018), and DaSiamRPN (Zhu et al., 2018). We identify combinations as either applicable ("A") or not ("N/A").

|  | ROMTrack | MixFormerM | TransT | DiMP | PrDiMP | SiamRPN | DaSiamRPN |
|---|---|---|---|---|---|---|---|
| SPARK(white-box) | N/A | N/A | A | N/A | N/A | A | A |
| RTAA (white-box) | N/A | N/A | A | N/A | N/A | A | A |
| IoU (black-box) | A | A | A | A | A | A | A |
| CSA (black-box) | A | A | A | N/A | N/A | A | A |

applicable on DiMP and PrDiMP trackers, as they do not have template and search regions in the tracking process. This lead us to achieve our experiments over two white-box attacks, SPARK and RTAA, and two black-box attacks, CSA and IoU.

# 4 Investigation

We conducted an analysis to determine how sensitive transformer trackers are to perturbations generated by existing attack methods under various conditions. We compared the difference in performance between the tracker's ability to provide accurate bounding boxes and binary masks by measuring the percentage difference from their original performance on clean data. We evaluated the impact of adversarial attacks on transformer trackers in predicting the object bounding boxes by varying the perturbation levels. Finally, we assessed the performance of the IoU attack when the generated perturbations were bounded at different noise levels. We then discussed the observations we drew from these sets of experiments.

## 4.1 Adversarial Attacks per Tracker Output

In this section, we have applied adversarial attack techniques against the TransT-SEG (Chen et al., 2023) and MixFormerM (Cui et al., 2022) trackers and compared the results based on different tracking outputs. The objective of this experiment is to determine the difference in each tracking metric before and after the attack when one of the tracker's outputs (bounding box or binary mask) is measured.

**Evaluation Protocol** We selected the VOT2022 Short-term dataset and protocol (Kristan et al., 2023) because of these three reasons. Unlike other datasets that use the one-pass evaluation protocol, the VOT2022-ST follows the anchor-based short-term protocol for the trackers evaluation. The metrics of VOT2022 baseline are obtained from the anchor-based short-term protocol which presents another view of the trackers' performances. Also, the VOT2022-ST provides the evaluation based on both object bounding boxes (STB) and binary masks (STS) which makes it a fair setup for our experiment. Therefore, the experiments for different attacks are achievable offline. Besides, the newest trackers are annually assessed by the VOT community and the most robust trackers are listed and announced on different sub-challenges. For instance, the MixFormerM tracker (Cui et al., 2022) was among five top-ranked trackers for binary mask prediction and its other variant, MixFormerL won third place on the bounding box prediction sub-challenge. We have conducted a baseline experiment for the VOT2022 (Kristan et al., 2023) short-term sub-challenge in two cases: object bounding box (STB) vs. object masks (STS) for target annotation and tracking. The Expected Average Overlap (EAO), accuracy and the anchor-based robustness metrics are calculated in this experiment. The EAO computes the expected value of the prediction overlaps with the ground truth. The accuracy measures the average of overlaps between tracker prediction and the ground truth over a successful tracking period. The robustness is computed as the length of a successful tracking period over the length of the video sequence. The successful period is a period of tracking in which the overlap between the prediction and ground truth is always greater than the pre-defined threshold. The baseline metrics are computed based on the anchor-based protocol introduced in VOT2020 (Kristan et al., 2020). In every video sequence evaluation under this protocol, the evaluation toolkit will reinitialize the tracker from the next anchor of the

Table 2: Evaluation results of the TransT-SEG (Chen et al., 2023) tracker attacked by different methods on the VOT2022 (Kristan et al., 2023) Short-Term (ST) dataset and protocol for two stacks: bounding box prediction via bounding box annotations (STB) and binary mask prediction via binary mask annotations (STS). The "Clean" values are the original tracker performance without applying any attack.

| Stack | Method | EAO | | | Accuracy | | | Robustness | | |
|---|---|---|---|---|---|---|---|---|---|---|
| | | Clean | Attack | Drop | Clean | Attack | Drop | Clean | Attack | Drop |
| STB | CSA | 0.299 | 0.285 | 4.68% | 0.472 | 0.477 | -1.06% | 0.772 | 0.744 | 3.63% |
| | IoU | 0.299 | 0.231 | 22.74% | 0.472 | 0.495 | -4.87% | 0.772 | 0.569 | 26.29% |
| | RTAA | 0.299 | 0.058 | 83.28% | 0.472 | 0.431 | 8.69% | 0.772 | 0.157 | 79.66% |
| | SPARK | 0.299 | 0.012 | 95.99% | 0.472 | 0.244 | 48.30% | 0.772 | 0.051 | 93.39% |
| STS | CSA | 0.500 | 0.458 | 8.40% | 0.749 | 0.736 | 1.73% | 0.815 | 0.779 | 4.42% |
| | IoU | 0.500 | 0.334 | 33.20% | 0.749 | 0.710 | 5.21% | 0.815 | 0.588 | 27.85% |
| | RTAA | 0.500 | 0.067 | 86.60% | 0.749 | 0.533 | 28.84% | 0.815 | 0.146 | 82.08% |
| | SPARK | 0.500 | 0.011 | 97.80% | 0.749 | 0.266 | 64.48% | 0.815 | 0.042 | 94.84% |

data to compute the anchor-based metrics wherever the tracking failure happens. For visualization usage, we employed some video sequences from DAVIS2016 (Perazzi et al., 2016) dataset.

**Attacks Setting** Four adversarial attacks are employed in this experiment, namely CSA (Yan et al., 2020), IoU (Jia et al., 2021), SPARK (Guo et al., 2020), and RTAA (Jia et al., 2020). However, not all of the attacks are applicable to both trackers. The SPARK and RTAA attacks manipulate the object candidates list, which includes classification labels and/or regression targets. If the tracker does not infer the candidates in the output, these attacks cannot be applied on, such as MixFormerM (Cui et al., 2022) that only outputs the predicted bounding box. In this experiment, we generated the perturbations using SPARK and RTAA attacks, i.e. white-box attacks, against trackers. However, the perturbation of CSA (Yan et al., 2020) is created by two GANs and passing the image into the SiamseRPN++ (Li et al., 2019) tracker to generate the adversarial loss depending on the SiamseRPN++ loss. Therefore, the CSA is a transferred black-box attack for both TransT-SEG (Chen et al., 2023) and MixFormer (Cui et al., 2022) trackers. The IoU method (Jia et al., 2021) is also a black-box approach that can perturb the whole frame using the tracker prediction for several times.

**Results** The following is a summary of the results obtained from an experiment conducted on the VOT2022 (Kristan et al., 2023) dataset using the TransT-SEG tracker (Chen et al., 2023) after adversarial attacks. The results are shown in Table 2 for both STB and STS cases. The most powerful attack against TransT-SEG (Chen et al., 2023) in all three metrics was found to be SPARK (Guo et al., 2020). According to the accuracy scores of Table 2, the object binary mask was more affected by the adversarial attacks than the object bounding box. However, the difference of drop percentages in assessing the predicted bounding boxes and binary masks are negligible for the EAO and Robustness metrics of TransT-SEG tracker.

It was observed that the CSA (Yan et al., 2020) attacks poorly degraded the outputs in evaluation metrics, except for the accuracy of the STB case. However, it was surprising to note that there was a negative difference in the accuracy metric after the CSA and IoU attack. Specifically, in the STB case, after the adversarial attacks, the accuracy of TransT-SEG (Chen et al., 2023) decreased by $-1.06\%$ for the CSA attack and $-4.87\%$ for the IoU attack. This improvement was also observed for MixFormer (Cui et al., 2022) in Table 3 for EAO and robustness metrics after the CSA attack in the STB case and accuracy after the CSA in the STS case of the experiments. The binary masks predicted by the MixFormerM tracker after the IoU attack for all computed metrics dropped greater than the metrics computed for the predicted bounding boxes. Nonetheless, following the CSA attack, the drop percentages of metrics for MixFormerM outputs were smaller and more consistent compared to those resulting from the IoU attack. Comparing the IoU attack results on both Tables 2 and 3 for bounding box evaluation (STB), the MixFormerM shows

Table 3: Evaluation results of the MixFormerM (Cui et al., 2022) tracker attacked by different methods on the VOT2022 (Kristan et al., 2023) Short-Term (ST) dataset and protocol for two stacks: bounding box prediction via bounding box annotations (STB), binary mask prediction via binary mask annotations (STS). The "Clean" values are the original tracker performance without applying any attack.

| Stack | Method | EAO | | | Accuracy | | | Robustness | | |
|---|---|---|---|---|---|---|---|---|---|---|
| | | Clean | Attack | Drop | Clean | Attack | Drop | Clean | Attack | Drop |
| STB | CSA | 0.303 | 0.308 | -1.65% | 0.479 | 0.478 | 0.21% | 0.780 | 0.791 | -1.41% |
| | IoU | 0.303 | 0.246 | 18.81% | 0.479 | 0.458 | 4.38% | 0.780 | 0.665 | 14.74% |
| STS | CSA | 0.589 | 0.562 | 4.58% | 0.798 | 0.803 | -0.63% | 0.880 | 0.857 | 2.61% |
| | IoU | 0.589 | 0.359 | 39.05% | 0.798 | 0.660 | 17.30% | 0.880 | 0.677 | 23.07% |

greater adversarial robustness with 14.74% drop percentage from original score whereas the robustness of TransT dropped by 26.29% after the same attack.

According to Table 3, the most powerful attack against MixFormerM (Cui et al., 2022) is the IoU attack (Jia et al., 2021). Even after the IoU attack (Jia et al., 2021), the adversarial perturbation slightly improves accuracy in assessing object binary masks (STS) and EAO in assessing predicted bounding boxes (STB). The EAO metric in the evaluation of the bounding box and binary mask is the most affected metric with 18.81% and 39.05% drop percentages after IoU attack. When compared to the corresponding metric for TransT-SEG (Chen et al., 2023) as shown in Table 2, the IoU attack had a greater damage on binary mask creation for MixFormerM than for TransT-SEG per EAO and accuracy metrics. However, this result was reversed for the same metrics when the object bounding box was evaluated in both trackers, Tables 2 and 3 (STB). For this point, it is important to mention that tracking and segmentation is performed by two different networks in MixFormerM (Cui et al., 2022). Therefore, the object bounding box evaluation is the assessment of the tracker's network while the binary mask evaluation is the assessment of the segmentation network (Yan et al., 2021). In contrast, the TransT-SEG (Chen et al., 2023) tracker performs both tracking and segmentation by a single transformer tracker.

Figure 1 demonstrates the results of different attacks on MixFormerM and TransT-SEG trackers in terms of the object bounding boxes and binary masks. The original outputs, bounding boxes and masks, are depicted by Green color, while the results after the attacks are exhibited with Red color. As the quantitative results indicated in Tables 2 and 3, the white-box attacks (SPARK and RTAA) have harmed the binary mask and bounding box more than the black-box attacks (IoU and CSA).

## 4.2 Adversarial Attacks per Perturbation Level

We test the effect of the perturbation levels on adversarial attack performance against transformer trackers. In white-box attacks such as SPARK (Guo et al., 2020) and RTAA (Jia et al., 2020), the generated perturbation is used to update the frame patch in each attack step. The overview of pseudocode of these two attacks is presented in Algorithms 1 and 2 where the $\phi_\epsilon$ indicates the operation of clipping the frame patch to the $\epsilon$-ball. The $\alpha$ is the applied norm of gradients, and $I$ is the search region. Although there are several differences in both settings, there is one similar step to generate the adversarial region from the input image gradient and previous perturbation(s). Line 4 of the RTAA pseudocode and line 6 of the SPARK pseudocode change the image pixels based on the computed gradients. By adjusting the $\epsilon$-ball, the performance of attacks is evaluated to demonstrate the power of each adversarial idea for object trackers. The plus or minus sign of the gradient sign corresponds to the decoupling direction and norm research in the gradient-based adversarial attack (Rony et al., 2019). For instance, SPARK (Guo et al., 2020) uses minus, while RTAA (Jia et al., 2020) sums up the sign of gradients with image values. Furthermore, note that in the original papers of SPARK (Guo et al., 2020) and RTAA (Jia et al., 2020), the attack parameters may have different names than those we used in this paper. Our goal was to unify their codes and approach into principle steps that make comparison more accessible for the audience. An important aspect of the

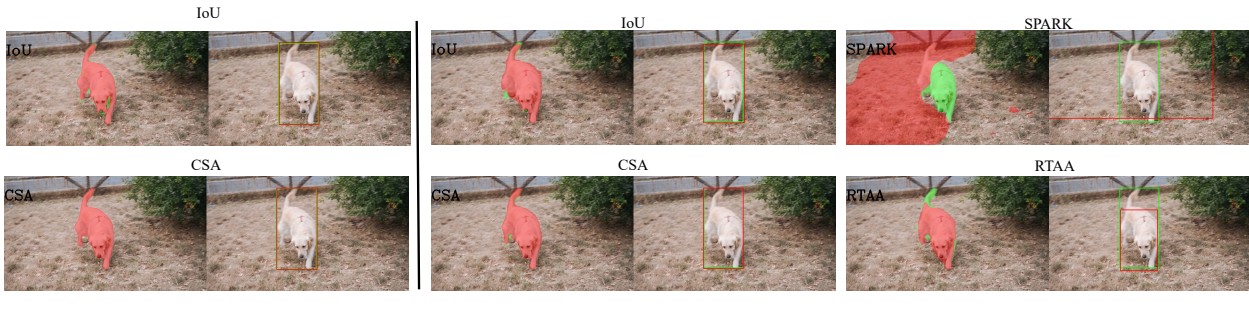

Figure 1: Mask vs. bounding box predictions as the output of transformer trackers, MixFormerM (Cui et al., 2022) and TransT-SEG (Chen et al., 2023), while the adversarial attacks applied to perturb the input frame/search region. The TransT-SEG tracker's outputs harmed by the white-box methods, SPARK (Guo et al., 2020) and RTAA (Jia et al., 2020), more than black-box attacks, IoU (Jia et al., 2021) and CSA (Yan et al., 2020). The green mask/bounding box represents the object tracker's performance while the red mask/bound box belongs to the tracker's performance after each attack.

---

**Algorithm 1** RTAA (Jia et al., 2020) algorithm as the adversarial attack for object trackers

---

1: $\mathcal{P} \leftarrow \mathcal{P}(t-1)$      ▷ Initialize with perturbation map of previous frame
2: $I^{\mathrm{adv}} \leftarrow I$      ▷ Initialize with clean current frame
3: **for** $i = 1, \ldots, i^{\max}$ **do**
4:      $I^{\mathrm{adv}} \leftarrow I^{\mathrm{adv}} + \phi^{\epsilon}\left(\mathcal{P} + \alpha \operatorname{sign}(\nabla_{I^{\mathrm{adv}}} \mathcal{L})\right)$      ▷ Application of adversarial gradient descent
5:      $I^{\mathrm{adv}} \leftarrow \max(0, \min(I^{\mathrm{adv}}, 255))$      ▷ Clamp image values in [0, 255]
6:      $\mathcal{P} \leftarrow I^{\mathrm{adv}} - I$      ▷ Update perturbation map
7: **Return** $I^{\mathrm{adv}}, \mathcal{P}$      ▷ Return adversarial image and corresponding perturbation map

---

SPARK algorithm, mentioned in (Guo et al., 2020), is its regularization term. This feature is convenient for maintaining sparse and imperceptible perturbations (Guo et al., 2020). The regularization term involves adding the $L_{2,1}$ Norm of previous perturbations to the adversarial loss, which helps generate sparse and imperceptible noises. We generated examples of SPARK perturbation (Guo et al., 2020) versus RTAA (Jia et al., 2020) perturbations to verify this claim.

**Evaluation Protocol** The test sets of the experiments on the perturbation level changes are the UAV123 dataset (Mueller et al., 2016) and VOTST2022 (Kristan et al., 2023). The UAV123 dataset comprises 123 video sequences with natural and synthetic frames in which an object appears and disappears from the frame captured by a moving camera. We calculate success and precision rates across various thresholds under the

---

**Algorithm 2** SPARK (Guo et al., 2020) algorithm as the adversarial attack for object trackers

---

1: $\mathcal{P} \leftarrow \mathcal{P}(t-1)$      ▷ Initialize with perturbation map of previous frame
2: $\mathcal{S} \leftarrow \sum_{i=1}^{K} \mathcal{P}(t-i)$      ▷ Sum of perturbation maps of last $K$ frames
3: $I^{\mathrm{adv}} \leftarrow I$      ▷ Initialize with clean current frame image
4: **for** $i = 1, \ldots, i^{\max}$ **do**
5:      $I' \leftarrow I^{\mathrm{adv}}$      ▷ Get a copy of current adversarial image
6:      $I^{\mathrm{adv}} \leftarrow I' + \phi^{\epsilon}\left(\mathcal{P} - \alpha \operatorname{sign}(\nabla_{I'} \mathcal{L})\right) + \mathcal{S}$      ▷ Application of adversarial gradient descent
7:      $I^{\mathrm{adv}} \leftarrow \max(0, \min(I^{\mathrm{adv}}, 255))$      ▷ Clamp image values in [0, 255]
8:      $\mathcal{P} \leftarrow I^{\mathrm{adv}} - I' - \mathcal{S}$      ▷ Update perturbation map
9: $\mathcal{S} \leftarrow \sum_{i=1}^{K} \mathcal{P}(t-i)$      ▷ Update the sum of perturbation maps
10: $I^{\mathrm{adv}} \leftarrow I + \mathcal{S}$      ▷ Generate the current adversarial frame
11: **Return** $I^{\mathrm{adv}}, \mathcal{P}$      ▷ Return adversarial image and corresponding perturbation map

---

One Pass Evaluation (OPE) protocol. In this setup, the object tracker is initialized using the first frame and the corresponding bounding box. Subsequently, the tracker is evaluated for each frame's prediction for the rest of the video sequence. Precision is measured by calculating the distance between the center of the ground truth's bounding box and the predicted bounding box. The precision plot shows the percentage of bounding boxes that fall within a given threshold distance. The success rate is computed based on the Intersection over Union (IoU) between the ground truth and predicted bounding boxes. The success plot is generated by considering different thresholds over IoU and computing the percentage of bounding boxes that pass the given threshold. Furthermore, we computed the L1 norm and structural similarity (SSIM) (Wang et al., 2004) as the measurements of sparsity and imperceptibility of the generated perturbations per attack. We chose some frames of the VOT2022ST (Kristan et al., 2023) dataset to visualize these metrics per frame.

**Attacks Setting**   The SPARK (Guo et al., 2020) and RTAA (Jia et al., 2020) approaches applied on the TransT tracker (Chen et al., 2021) are assessed in this experiment using the OPE protocol. We chose the TransT tracker which is a pioneer on transformer trackers to observe the attack performance change on the perturbation levels. Both attacks generate the perturbed search region over a fixed number of iterations (10). While the step size $\alpha$ for the gradient's update is 1 for RTAA and 0.3 for SPARK. We used five levels of perturbation $\epsilon \in \{2.55, 5.1, 10.2, 20.4, 40.8\}$ to compare its effects on the TransT (Chen et al., 2021) performance on UAV123 (Mueller et al., 2016) and VOT2022ST (Kristan et al., 2023) datasets. The $\epsilon$'s are selected as a set of coefficients $\{0.01, 0.02, 0.04, 0.08, 0.16\}$ of the maximum pixel value 255 in an RGB image. It is worth mentioning that the $\epsilon$ for both attacks are set to 10 in their original settings. Therefore, the original performance of each attack is very close to the $\epsilon_3 = 10.2$ perturbation level.

**Results**   Figure 2 shows the performance of TransT (Chen et al., 2021) under RTAA (Jia et al., 2020) and SPARK (Guo et al., 2020) attacks with different perturbation levels. The red curve indicates the clean performance of the tracker before applying any attacks. The other perturbation levels are demonstrated with different colors. Unlike classification networks with transformer backbones (Shao et al., 2022), the transformer tracker performances after the RTAA attack (Jia et al., 2020) using different $\epsilon$'s are minimally different but not after SPARK attacks (Guo et al., 2020). Adversarial perturbation methods against trackers use the previous perturbation to be added to the current frame. This setting may remove the sensitivity of the attack methods in the perturbation levels. In the RTAA attack (Jia et al., 2020), only one last perturbation is added to the current frame. In contrast, the SPARK (Guo et al., 2020) uses the previous perturbations in each time step for the last $K = 30$ frames, which reduces the sensitivity of the output to small changes in the inputs. For perturbation levels $\{\epsilon_3, \epsilon_4, \epsilon_5\}$, RTAA's performance (Jia et al., 2020) remains the same, whereas using smaller levels affects its performance. It is noteworthy that RTAA (Jia et al., 2020) outperforms SPARK (Guo et al., 2020) on UAV datasets (Mueller et al., 2016) in almost every perturbation level except for the most minor level $\epsilon_1 = 0.01$ in which SPARK is the stronger attack.

In the main paper of SPARK (Guo et al., 2020), it has been mentioned that the technique generates a temporally sparse and imperceptible noise. Figure 3 displays various examples of perturbed search regions and perturbation maps produced by the TransT tracker after applying the SPARK attack. Upon applying the attack, we noted that some frames, like frame number "7" of the 'bubble' sequence and the first two rows of Figure 3, generated search regions and perturbation maps with fixed values for imperceptibility (SSIM metric) and sparsity (L1 norm). Even by increasing the perturbation level, some frames retained the same level of imperceptibility and sparsity. However, there were also instances of super-perturbed search regions per video sequence, where the noise was noticeable, and the L1 norm had a high value, as shown in the last two rows of Figure 3. We consider a perturbed search region super-perturbed when the imperceptibility of the region is lower than 50%. The SPARK algorithm generates the most imperceptible noise with a constant and high SSIM value of 99.95% and sparse noise with L1 norm of 40.96 in all of the perturbation levels given the same frame. This fixed number of L1 norm is also the result of the regularization term discussed in the SPARK paper (Guo et al., 2020). This stability of SSIM and L1 norm have been repeated for many frames of the "bubble" sequence for SPARK attack (Guo et al., 2020) while in some frames, the imperceptibility and sparsity are not stable per perturbation levels.

In Figure 3, we indicate some super-perturbed regions with their perturbation maps per perturbation level. Interestingly, as we increase the perturbation levels, the number of super-perturbed regions also increases. In the attack settings, the perturbation of previous frames considered in the loss function is erased every 30

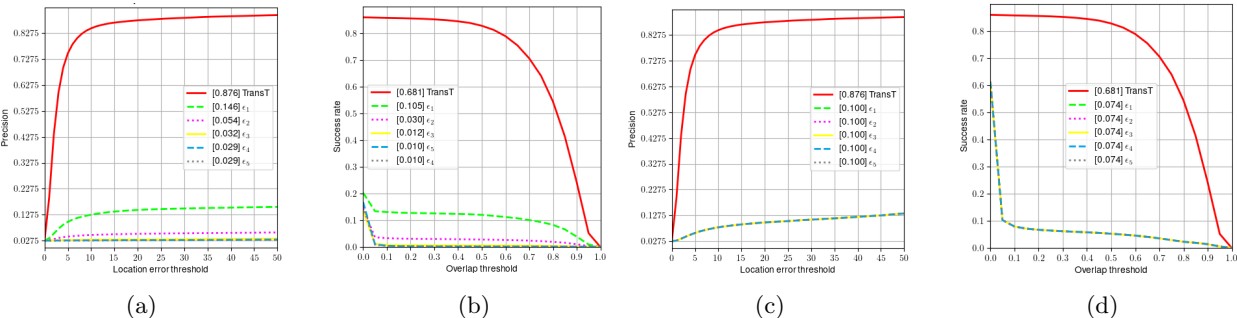

Figure 2: The precision and success plots related to the TransT (Chen et al., 2021) performance after RTAA (Jia et al., 2020) (a, b) and SPARK (Guo et al., 2020) (c,d) attack under different levels of noise on UAV123 (Mueller et al., 2016) dataset. The average score for each metric is shown in the legend of the plots. The 'red' plot is the original TransT performance without any attack applied on the tracker. The $e$'s are corresponded to $\epsilon$'s in our experiment, changing from $e_1 = 2.55$ to $e_5 = 40.8$ to assess the TransT performance after the white-box attacks under various perturbation levels. The SPARK performances per perturbation level shifts did not change on UAV123dataset as one can observe the SPARK curves are overlapped.

Table 4: The perturbation levels versus number of highly perturbed search regions generated by the SPARK algorithm (Guo et al., 2020) applied on TransT (Chen et al., 2021) tracker. The SSIM and L1 norm are computed as the average number of highly perturbed regions on the "bubble" sequence of VOT2022 (Kristan et al., 2023) dataset. The "No. of frames" is the number of super-perturbed frames in which the SSIM value is below than 50%.

| $\epsilon$ | No. of frames | SSIM | L1 norm |
|---|---|---|---|
| 2.55 | 7 | 36.86 | 176.04 |
| 5.1 | 7 | 40.96 | 181.86 |
| 10.2 | 13 | 41.08 | 181.33 |
| 20.4 | 13 | 41.97 | 182.53 |
| 40.8 | 14 | 42.53 | 183.98 |

frames for RTAA (Jia et al., 2020) and SPARK (Guo et al., 2020) algorithms. Table 4 provides information about the number of highly perturbed frames during a video sequence and the average imperceptibility (SSIM) and sparsity (L1 norm) scores. With higher levels of perturbations, more frames become highly perturbed, resulting in a greater L1 norm of the perturbations. Furthermore, in the lower perturbations levels, the highly perturbed search regions generate more perceptible noise, i.e. the imperceptibility of generated perturbations have grown by boosting the perturbation level.

For the RTAA (Jia et al., 2020) attack applied on the TransT (Chen et al., 2021) tracker, whenever the perturbation level boosts the imperceptibility and sparsity declines. Figure 4 demonstrates the result of applying RTAA against TransT tracker for the same frame #7 of the 'bubble' sequence. The RTAA attack perturbs search regions with higher SSIM values at the lowest $\epsilon$ level, i.e. the first level $\epsilon = 2.55$. By increasing the perturbation levels, the perceptibly of the RTAA perturbation has been increased while the sparsity changes are small.

## 4.3 Adversarial Attack per Upper-Bound

We conducted an experiment to test the vulnerability of the IoU attack (Jia et al., 2021) to noise bounding, using different upper bounds. The IoU method (Jia et al., 2021) is a black-box attack that misleads trackers by adding various noises to the frame using object bounding boxes. The essential steps of the IoU attack (Jia et al., 2021) involve creating two levels of noise perturbations: orthogonal and normal direction noises. Our

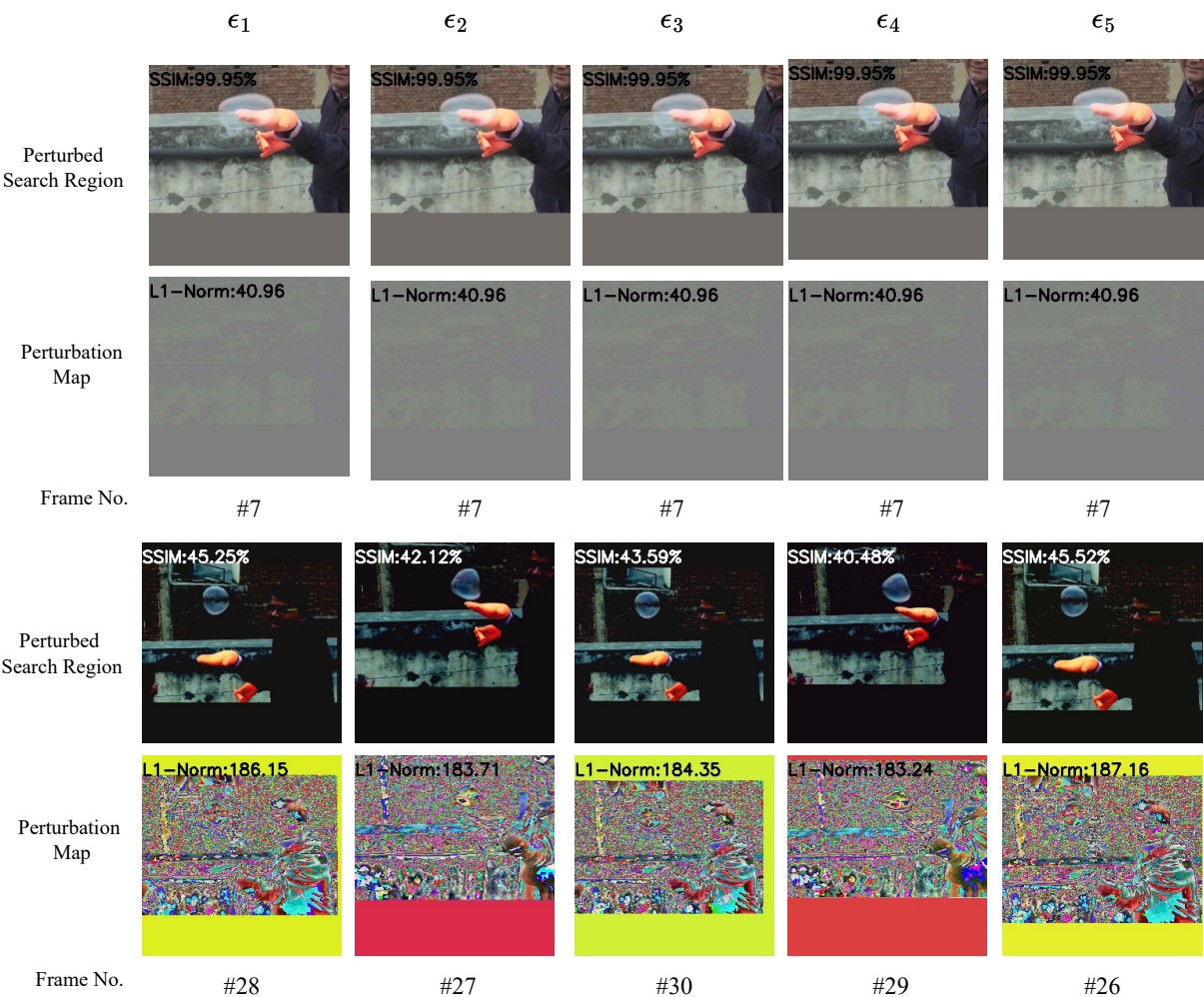

Figure 3: The search regions related to the "bubble" sequence in the VOT2022ST dataset (Kristan et al., 2023) after applying SPARK (Guo et al., 2020) attack on TransT (Chen et al., 2021) tracker. The perturbed search region is labeled with the SSIM (Wang et al., 2004) measured between search regions before and after the attack. The perturbation maps, following the work of (Yan et al., 2020), are created to demonstrate the added noise in colors. The L1 norm for perturbation maps are calculated to show the perturbation density/sparsity.

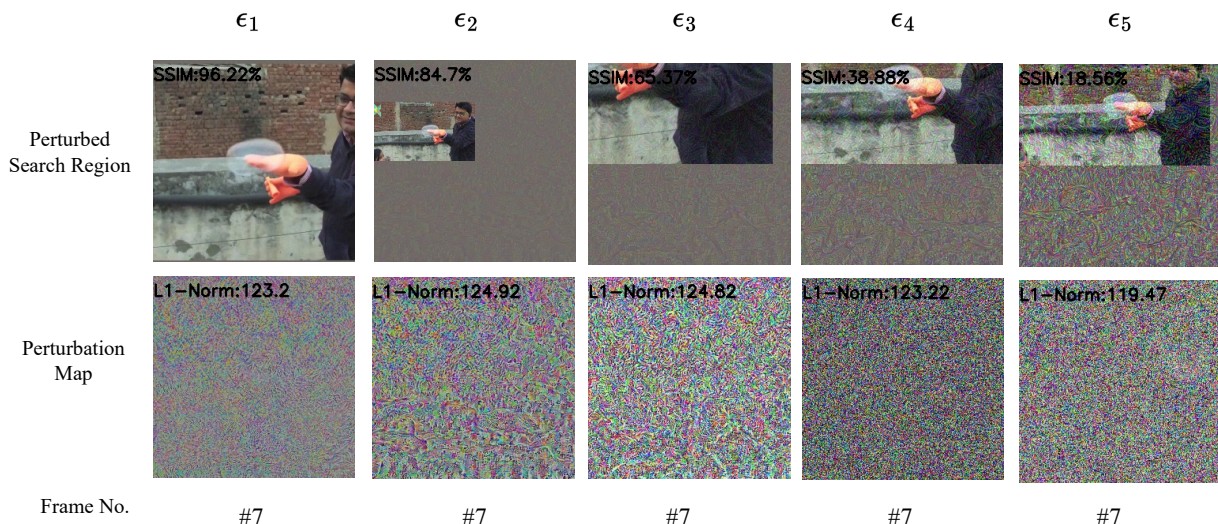

Figure 4: The search regions related to the "bubble" sequence in the VOT2022ST dataset (Kristan et al., 2023) after applying RTAA (Jia et al., 2020) attack on TransT (Chen et al., 2021) tracker. The perturbed search region is labeled with the SSIM (Wang et al., 2004) measured between search regions before and after the attack. The perturbation maps, following the work of (Yan et al., 2020), are created to demonstrate the added noise in colors. The L1 norm for perturbation maps are calculated to show the perturbation density (i.e. sparsity).

study aims to manipulate the attack settings in the second part of perturbation generation, which is in the normal direction.

**Dataset and Protocol**    The performances of the IoU attack are assessed against ROMTrack (Cai et al., 2023) on UAV123 (Mueller et al., 2016) dataset using the OPE protocol. The success rate, precision rate, and normalized precision rate are computed to compare the results. For this experiment, we report the average of the success rate called Area Under Curve (AUC), as well as the average precision and norm precision on the thresholds.

**Attack Setting**    The IoU method (Jia et al., 2021) is a black-box attack on object trackers. It adds two types of noise to the frames: one in the tangential direction and the other in the normal direction. In the original setting, there was no limit on the number of times that noise could be added in the normal direction. This noise was limited only by the upper bound $\zeta$ and the $S_{IoU}$ value in the original setting. The $S_{IoU}$ value is the weighted average of two computed IoU values: 1) the IoU of the noisy frame prediction and the first initialized frame $I_1^{adv}$ in the attack algorithm, and 2) the IoU of the noisy frame prediction and the last frame bounding box in the tracking loop. In our experiment, we set a limit of 10 steps in the algorithm's last loop to reduce the processing time, especially for the larger upper bound $\zeta$ values. We tested the IoU attack under three upper bounds: $\zeta \in \{8000, 10000, 12000\}$. The middle value of $\zeta = 10000$ corresponds to the original setting of the IoU attack (Jia et al., 2021).

**Results**    The images in Figure 5 display the results of the IoU attack (Jia et al., 2021) against ROMTrack (Cai et al., 2023) under various upper bounds for a single frame. The L1 norm of the perturbation has increased as the upper bounds were raised. Additionally, the imperceptibility, measured by the SSIM values, decreased as the perturbations became more severe. Since the IoU attack starts by generating some random noise, it is highly dependent on the initialization points. For some cases, the algorithm did not process a single video sequence even after 48 hours. One solution that worked for proceeding was to stop the processing without saving any results about the current sequence to restart the evaluation. After re-initialization, the attack began from another random point (noise) and it proceeded to the next sequence in less than 2 hours.

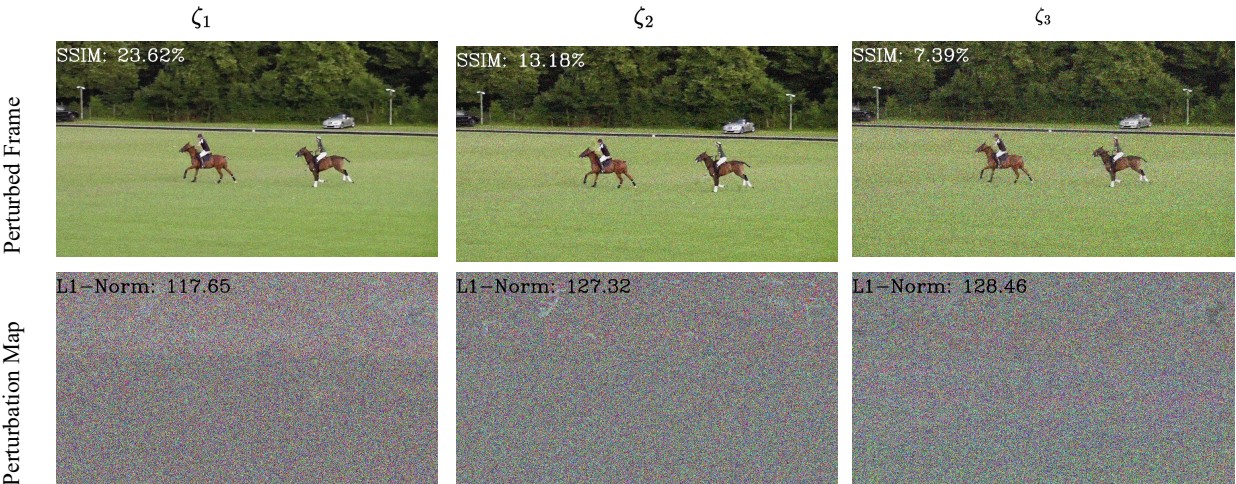

Figure 5: The perturbed frames and perturbation maps generated by the IoU method (Jia et al., 2021) against ROMTrack (Cai et al., 2023) using three upper bounds of $\zeta \in \{8k, 10k, 12k\}$. The imperceptibility and L1 norm of the generated perturbations are shown in the frames representing the noise imperceptibility and sparsity of perturbation maps.

Table 5: Evaluation results of the ROMTrack (Cai et al., 2023) attacked by the IoU approach (Jia et al., 2021) on the UAV123 (Mueller et al., 2016) dataset and protocol for three different upper bounds on the added noise in normal direction up to 10 processing steps.

| | AUC | | | Precision | | | Norm Precision | | |
|---|---|---|---|---|---|---|---|---|---|
| $\zeta$ | Original | Attack | Drop | Original | Attack | Drop | Original | Attack | Drop |
| 8k | 69.74 | 66.85 | 4.14% | 90.83 | 89.31 | 1.67% | 85.30 | 83.00 | 2.70% |
| 10k | 69.74 | 65.46 | 6.14% | 90.83 | 87.81 | 3.32% | 85.30 | 81.73 | 4.18% |
| 12k | 69.74 | 63.61 | 8.79% | 90.83 | 86.31 | 4.98% | 85.30 | 79.71 | 6.55% |

The results of the attack on ROMTrack (Cai et al., 2023) using the IoU method (Jia et al., 2021) with different upper bounds are presented in Table 5. It is clear that a higher upper bound leads to a more effective attack across all metrics. Despite the most substantial level of perturbation using the IoU method (Jia et al., 2021) resulting in an 8.79% decrease in the AUC metric, this outcome is insignificant. As shown in Figure 5, increasing $\zeta$ generates a perceptible perturbation with a lower SSIM to the original frame, resulting in a noisier frame that damages the tracking performance of ROMTrack (Cai et al., 2023) even more. However, the robust tracking performance is not affected more than 9% per metric, even in the highest perturbation level. In other words, ROMTrack (Cai et al., 2023) demonstrates good adversarial robustness against IoU attack on UAV123 dataset.

## 4.4 Transformer versus Non-transformer Trackers

In this experiment, we aim to study the adversarial robustness of trackers with different backbones especially transformer-based trackers compared to the non-transformer trackers.

**Evaluation Protocol**  This experiment is performed on the GOT10k dataset (Huang et al., 2019) which uses OPE protocol and the results are reported for three metrics: Average Overlap (AO), Success Rate $SR_{0.5}$ with threshold 0.5 and Success Rate $SR_{0.75}$ with threshold 0.75. The GOT10k test set contains 180 video sequences with axis aligned bounding box annotations.

**Attacks Setting**  We applied a set of attacks including two white-box attacks, RTAA (Jia et al., 2020), SPARK (Guo et al., 2020), and two black-box attacks, IoU (Jia et al., 2021) and CSA (Yan et al., 2020), against a set of trackers including three transformer trackers (i.e., TransT (Chen et al., 2021), MixFormer (Cui et al., 2022) and ROMTrack (Cai et al., 2023)), two Siamese-based trackers (i.e., SiamRPN (Li et al., 2018) and DaSiamRPN (Zhu et al., 2018)), and two discriminative trackers (i.e., DiMP (Bhat et al., 2019) and PrDiMP (Danelljan et al., 2020)). As it is discussed in Attack Setups Section 3.3 and Table 1, some attack methods are not applicable to some trackers due to the unavailability of the input for the attack algorithms. The RTAA and SPARK are used as a form of white-box attack, while the IoU and CSA methods are black-box attacks. For MixFormer and ROMTrack trackers, the related pre-trained networks specialized for GOT10k dataset and their parameters are loaded and examined. For DaSiamesRPN tracker, the checkpoint released to test the OTB100 (Wu et al., 2015) dataset is used in this experiment.

**Results**  Table 6 indicates the vulnerability of the object trackers with different backbones, transformers vs. non-transformers, against the adversarial attacks. The tracking performance before the attacks is indicated as the "No Attack" scores for each tracker, while the evaluation metrics after the attack are shown under the attack names. The "Drop(%)" columns in Table 6 represents the drop percentage of scores from the original tracker response per metrics, and it is calculated as $(100 \times [\text{original} - \text{after attack}]/\text{original})$. The drop scores are calculated to demonstrate the attack performances in comparison to each other and to assess the tracker's performance after each attack compared to other trackers.

Beginning with the transformer trackers, the "No Attack" scores exhibit a relatively higher magnitude compared to non-transformer trackers. Within transformer trackers, the IoU and CSA methods are the mutual attacks, with the IoU method demonstrating a more significant damage compared to the CSA approach. Specifically, among transformer trackers, TransT appears to be particularly susceptible to perturbations induced by the IoU attack. However, a greater decline in performance for MixFormer following the CSA attack is observed in compared to other transformer trackers after the same attack, i.e. CSA attack.

All the test attack approaches could be applied to TransT, SiamRPN, and DaSiamRPN trackers due to the availability of the classification and regression labels in these trackers. Intriguingly, the RTAA method provides the most powerful attacking performance against SiamRPN and DaSiamRPN and TransT trackers. Although the drop percentages of scores after SPARK attack is very close for Siamese-based trackers, this attack performance on TransT tracker is considerable. Repeated on for Siamese-based trackers, the IoU attack is resulted in greater performance drop rather than the CSA attack.

Another intriguing observation is that MixFormer shows the smallest drop percentage after the IoU attack among transformer trackers. For example, the $SR_{0.5}$ of MixFormer drops by 9.86%, whereas the percentage drop of the same metric is 18.31% for ROMTrack and 25.76% for TransT tracker. This indicates that

Table 6: The performance of transformer (Chen et al., 2023; Cui et al., 2022; Cai et al., 2023) and non-transformer trackers (Li et al., 2018; Zhu et al., 2018; Bhat et al., 2019; Danelljan et al., 2020) after white-box, SPARK (Guo et al., 2020) and RTAA (Jia et al., 2020), and black-box, IoU (Jia et al., 2021) and CSA (Yan et al., 2020), attacks on the GOT10k (Huang et al., 2019) dataset.

| Tracker | Attacker | Scores | | | Drop(%) | | |
|---|---|---|---|---|---|---|---|
| | | AO | $SR_{0.5}$ | $SR_{0.75}$ | AO | $SR_{0.5}$ | $SR_{0.75}$ |
| ROMTrack | No Attack | 0.729 | 0.830 | 0.702 | - | - | - |
| | CSA | 0.716 | 0.814 | 0.682 | 1.78 | 1.93 | 2.85 |
| | **IoU** | 0.597 | 0.678 | 0.536 | **18.11** | **18.31** | **23.65** |
| TransT | No Attack | 0.723 | 0.823 | 0.682 | - | - | - |
| | CSA | 0.679 | 0.768 | 0.628 | 6.08 | 6.68 | 7.92 |
| | IoU | 0.532 | 0.611 | 0.432 | 26.42 | 25.76 | 36.66 |
| | SPARK | 0.137 | 0.085 | 0.032 | 81.05 | 89.67 | 95.31 |
| | **RTAA** | 0.048 | 0.019 | 0.011 | **93.36** | **97.69** | **98.39** |
| MixFormer | No Attack | 0.696 | 0.791 | 0.656 | - | - | - |
| | CSA | 0.638 | 0.727 | 0.572 | 8.33 | 8.10 | 12.80 |
| | **IoU** | 0.625 | 0.713 | 0.543 | **10.20** | **9.86** | **17.22** |
| PrDiMP | No Attack | 0.645 | 0.751 | 0.540 | - | - | - |
| | **IoU** | 0.585 | 0.696 | 0.421 | **9.30** | **7.32** | **22.04** |
| DiMP | No Attack | 0.602 | 0.717 | 0.463 | - | - | - |
| | **IoU** | 0.549 | 0.653 | 0.372 | **8.80** | **8.93** | **19.65** |
| SiamRPN | No Attack | 0.406 | 0.499 | 0.101 | - | - | - |
| | CSA | 0.382 | 0.460 | 0.083 | 5.91 | 7.81 | 17.82 |
| | IoU | 0.364 | 0.442 | 0.107 | 10.34 | 11.42 | -5.94 |
| | SPARK | 0.279 | 0.353 | 0.059 | 31.28 | 29.26 | 41.58 |
| | **RTAA** | 0.032 | 0.013 | 0.001 | **92.12** | **97.39** | **99.00** |
| DaSiamRPN | No Attack | 0.389 | 0.465 | 0.090 | - | - | - |
| | CSA | 0.314 | 0.353 | 0.061 | 19.28 | 24.09 | 32.22 |
| | IoU | 0.320 | 0.376 | 0.081 | 17.74 | 19.14 | 10 |
| | SPARK | 0.255 | 0.313 | 0.053 | 34.44 | 32.69 | 41.11 |
| | **RTAA** | 0.037 | 0.015 | 0.001 | **90.49** | **96.77** | **98.88** |

although ROMTrack and TransT initially achieve better scores before the attacks, their performance is more greatly affected by IoU attack, resulting in larger drops in their evaluation metrics. Respectively, ROMTrack, TransT and MixFormer are ranked in terms of adversarial robustness after both IoU and CSA attack per AO and $SR_{0.5}$ metrics among all trackers. However, for $SR_{0.75}$ measurements, the ranking of adversarial robustness is MixFormer, ROMTrack and TransT after IoU attack.

Conversely, the only applicable attack (from our attack methods) against discriminative trackers, DiMP and PrDiMP, is the black-box IoU method. A comparison between these two trackers reveals that PrDiMP exhibits greater robustness than DiMP. However, upon examining the evaluation scores, it's evident that PrDiMP experiences a more pronounced decline compared to DiMP in terms of percentage drop of $SR_{0.75}$ metric. As the percentage drop related to $SR_{0.75}$ for PrDiMP is 22.04%, while it is 19.65% for the same score of DiMP tracker. For the drop percentage of $SR_{0.5}$ score, the PrDiMP, though, preserve its priority over DiMP tracker after the IoU attack.

The TransT, ROMTrack and DaSiamRPN are the top three in drop percentages of AO and $SR_{0.5}$ scores after the IoU attack. The computed percentage drops indicates that although the transformer trackers demonstrate the highest scores before the attack, their scores after IoU attack fall more significantly than the non-transformer trackers in general. In some cases such as DaSiamRPN tracker, the percentage drop is also a big number per AO and $SR_{0.5}$ metrics.

## 5 Discussion

Our investigation began in Section 4.1 with an examination of the adversarial robustness of transformer trackers in producing object bounding boxes compared to object binary masks. Our findings indicated that the prediction of binary masks was more susceptible to adversarial perturbations than object bounding box predictions, particularly in terms of the accuracy metric. Additionally, we observed that white-box attacks, specifically SPARK and RTAA, exhibited greater efficacy compared to black-box attacks such as IoU and CSA, when targeting the TransT-SEG tracker. Notably, among the transformer-based trackers analyzed, MixFormerM, which employs deeper relation modeling than TransT-SEG, demonstrated superior adversarial robustness in terms of computed EAO, accuracy, and robustness on the VOT2022STS dataset against a single attack. Furthermore, we observed that MixFormerM is not susceptible to attacks like SPARK and RTAA with its own gradients due to the absence of necessary attack proxies, namely classification and regression labels.

In the subsequent experiment (c.f. Section 4.2), we demonstrated that the magnitude of perturbation shifts could influence or maintain the overall tracking outcomes, depending upon the attack strategy. An elevated level of perturbation consistently resulted in a higher count of super-perturbed search regions with increased L1 norm values for perturbations in both RTAA and SPARK. However, many perturbed search regions under the SPARK attack demonstrated the same values for SSIM and L1 norm metrics, indicating a stable imperceptibility and sparsity even with heightened levels of perturbation.

For the third experiment (c.f. Section 4.3), we evaluated the IoU attack across various upper bounds and discovered that, particularly in the context of black-box attacks involving random noises, the initialization point plays a crucial role. The IoU method becomes exceedingly time-consuming due to improper initialization. Higher upper bounds resulted in larger L1 norm values and smaller SSIM scores, suggesting less sparse and more perceptible noise in the perturbed frames.

In the concluding experiment (c.f. Section 4.4), we examined the impact of various backbones—transformer-based, discriminative-based and Siamese-based—on visual tracking before and after adversarial attacks. Despite transformer trackers (ROMTrack, TransT, and MixFormer) showcasing the top-3 performance, their evaluation scores more notably decreased after applying the IoU method.

Another interesting finding is that the most effective attack may vary depending on the test set and tracker. Indeed, for TransT-SEG, the strongest attack on the VOT2022 dataset was SPARK (Section 4.1), whereas RTAA outperformed SPARK on the GOT10k set on misleading TransT tracker's output (Section 4.4). For the SiamRPN tracker, the similar trend on the GOT10k set was observed: RTAA outperformed SPARK.

In our settings, the only applicable attacks against MixFormer and ROMTrack are black-box attacks, i.e. IoU and CSA. Among black-box methods, the IoU attack outperformed CSA for TransT, MixFormer, Mix-FormerM and ROMTrack trackers, Sections 4.1 and 4.4. However, the effect of the IoU method against ROMTrack and MixFormer's are trivial, Section 4.4. The ROMTrack and MixFormer bounding box predictions were harmed by the IoU method up to 18.11 % and 18.81% on GOT10k and VOT2022 datasets for the average overlap metric, respectively. This indicates that these trackers were not being challenged enough with existing applicable attack methods.

In addition to the aforementioned summary, our study also revealed the following observations:

- The generated perturbations for attacks that are applicable to transformer-based trackers have more impact on the object masks accuracy rather than on the accuracy of the bounding boxes on VOT2022ST dataset (Section 4.1).
- Although it was demonstrated that adding previous perturbations to the current frame for perturbed search regions generation have an impact on the attacker performance (Guo et al., 2020), these previous perturbations result in more stable performance against changes in perturbation levels. For instance, such stability has been observed on SPARK (Guo et al., 2020) attack performance against TransT tracker on the UAV123 dataset (Mueller et al., 2016) (Section 4.2).
- The SPARK algorithm generates temporally sparse perturbations, meaning that the added perturbation to the search region is small for many frames. It results in imperceptible noise for those frames per video sequence, even though the perturbation level shifts to a higher value (Section 4.2).
- Increasing the perturbation level on SPARK results in more super-perturbed regions, i.e. regions with perceptible noise (Section 4.2).
- In IoU attack approach (Jia et al., 2020) and RTAA (Jia et al., 2021) attack, adding a higher perturbation level generates more perceptible noise for all frames, which damage more the overall tracking performance (Sections 4.2 and 4.3).
- The ranking of attack performance is sensitive to the experiment settings, dataset and protocol. For instance, SPARK method outperforms RTAA attack on VOT2022 for TransT-SEG tracker in Section 4.1, while RTAA scores are smaller than SPARK scores for TransT tracker on UAV123 dataset (Section 4.2). Again we observed that RTAA outperforms SPARK approaches in attacking TransT, SiamRPN and DaSiamRPN trackers on GOT10k set (Section 4.4).
- The outcome of the IoU attack is sensitive to its initialization. The evaluation process may take a long time due to unsuitable initialization point of the attack (Section 4.3).
- Although transformer trackers, containing ROMTrack, MixFormer and TransT, exhibits more robust performance before and after the adversarial attacks, comparing the percentage drops from original scores reveals that these tracker's average overlap decreased from the original scores greater than almost all other trackers after the IoU attack (Section 4.4).
- Discriminative trackers also demonstrate a great adversarial robustness and ranked immediately after the transformer trackers on GOT10k dataset (Section 4.4).

# 6    Conclusion

We conducted a study on adversarial attack methods for object trackers with the aim of testing their impact on transformer trackers and comparing their influences on visual trackers with different backbones. This paper includes several experiments on various tracking datasets, trackers, and attack settings. We evaluated three transformer trackers, ranging from light to deep relation modeling, and four non-transformer trackers with Siamese-based and discriminative backbones. Four attack methods, including two in white-box and two in black-box settings, were employed to assess the adversarial robustness of visual trackers. Our results showed that the accuracy of binary masks are more likely to harm by the adversarial attack in comparison to the accuracy of predicted bounding boxes. We also discovered that changes in the perturbation level do not necessarily affect the tracking performance over a tracking dataset. The sparsity and imperceptibility of the perturbations can be managed by advising a proper loss function. We also found that transformer trackers' performances after the adversarial attack drops from original performance greatly when compared to Siamese-based and discriminative trackers after the same attack. Our study indicates the need for further research on the adversarial robustness of transformer trackers since the existing attacks did not challenge

these trackers significantly. The white-box attacks against Siamese-based trackers are not applicable to the transformer and discriminative trackers using their gradients due to the change of tracker backbone and architecture. One potential path for future work is the development of novel white-box attacks to target these kinds of trackers. The other direction can be on focusing more effective black-box attacks since they do not depend on the tracker backbones. The insightful findings in the adversarial robustness of classification networks with transformer backbones are also worthy to transfer on tracking networks with transformer backbones.

### Acknowledgments

This work is supported by the DEEL Project CRDPJ 537462-18 funded by the Natural Sciences and Engineering Research Council of Canada (NSERC) and the Consortium for Research and Innovation in Aerospace in Québec (CRIAQ), together with its industrial partners Thales Canada inc, Bell Textron Canada Limited, CAE inc and Bombardier inc. [1]

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
