# OpenReview forum: "Reproducibility Study on Adversarial Attacks Against Robust Transformer Trackers"
_TMLR — Accepted by TMLR_

### Review · Reviewer_ChwA · 2024-03-08

**Summary Of Contributions:**

This paper conducted an empirical study on applying existing adversarial attacks against object tracking models to Transformer-based tracking models.

**Audience:**

Yes

**Claims And Evidence:**

No

**Requested Changes:**

Critical:
1. I'd suggest authors add a comparison between models with different backbones (Transformer-based v.s. non-Transformer-based) with a detailed analysis.
2. I'd also suggest authors analyze how Transformer makes a difference here.

**Strengths And Weaknesses:**

Strengths:
1. The paper demonstrated a comprehensive empirical study on the robustness of different Transformer-based trackers against adversarial attacks.

Weaknesses:
1. This paper does not propose anything new beyond applying existing methods to models with a different backbone (Transformer).
2. The paper does not show how Transformer makes a difference in the problem (which is the major difference of this paper compared to existing works). A comparison and deep analysis on the adversarial robustness of models with different backbones (Transformer-based v.s. non-Transformer-based) is lacking. Currently the paper is limited to only showing the empirical results without further insights.

Minor comments:
1. On page 5, it is mentioned that "the object binary mask was more affected by the adversarial attacks than the object bounding box". But I think the difference looks small to me, according to Table 1.

---

> ### Author Response · Authors · 2024-04-21
> **Q1. Contributions**
>
> We thank the reviewer for their constructive feedback. We are glad to note that our manuscript demonstrates a comprehensive empirical study on the adversarial robustness of transformer trackers. We reply to each of the reviewer’s points below.
>
> **"A. This paper does not propose anything new beyond applying existing methods to models with a different backbone (Transformer).
> B. The paper does not show how Transformer makes a difference in the problem (which is the major difference of this paper compared to existing works). A comparison and deep analysis on the adversarial robustness of models with different backbones (Transformer-based v.s. non-Transformer-based) is lacking. Currently, the paper is limited to only showing the empirical results without further insights."**
>
> This revised version of the paper now includes a thorough comparison with non-transformer trackers including Siamese-based and discriminative trackers. Our analysis reveals that transformer trackers exhibit higher tracking scores regarding AUC and precision. However, their score percentage drops are greater when adversarially attacked, in comparison to other trackers.
>
> In Sec. 4.4, we explained the experiment of transformer and non-transformer trackers compared before and after the adversarial attacks applied to them.
> Sec. 1 has been updated to more clearly highlight our contribution to the field. As a reproducibility study, we provided a set of experiments to capture the tracking performance of novel transformer trackers against the adversarial attack which fills a gap between the new transformer trackers and the adversarial attacks for visual trackers in the literature. In addition to these reproducibility results, our manuscript includes two key novel contributions:
>
> 1. We extend the application of adversarial attacks, originally designed for non-transformer trackers like SPARK and RTAA, to assess their effectiveness against transformer-based trackers.
>
> 2. We thoroughly evaluate the adversarial robustness of transformer-based trackers across various output scenarios, perturbation levels, changes in upper bounds, and in comparison to the non-transformer trackers.

---

> ### Author Response · Authors · 2024-04-21
> **Q2. Tracker outputs**
>
> **“On page 5, it is mentioned that "the object binary mask was more affected by the adversarial attacks than the object bounding box". But I think the difference looks small to me, according to Table 1.”**
>
> We observe a noticeable difference between the performance on binary mask and bounding box on the accuracy metric. According to Table 1, TransT-SEG tracker scores after the IoU attack, the drop percentage of accuracy is reported as -4.87% in evaluating the predicted bounding boxes, while the drop percentage of accuracy is computed as 5.21% in evaluating the object binary masks. Therefore, there is a total difference of 10.08% in these two cases where the evaluation is based on the bounding boxes and binary masks for one tracker attacked by the same method. As another example, we observed that the drop percentage of accuracy after the SPARK attack applied on TransT-SEG is 8.69% for assessing the predicted bounding boxes and 28.84% for assessing the object binary masks. However, the reviewer is right in mentioning that the difference is negligible for the EAO and Robustness metrics of TransT-SEG.
>
> Furthermore, the MixFormer scores for all three metrics change significantly while the evaluation is performed per bounding box versus binary masks under IoU attack. For instance, the percentage drop of EAO after IoU attack is changed from 18.81% for predicted bounding boxes to 39.05% for the object binary masks. The percentage drop in accuracy of MixFormer after the IoU attack is 4.38% in assessing with object bounding boxes and 17.30% in assessing with the object binary masks. However, the reviewer is right in mentioning that the difference is trivial for MixFormer tracker after the CSA attack.
>
> The claims in Sec. 4.1 and Sec. 5 of the revised manuscript has been adjusted to reflect this observation:
>
> “The generated perturbations for attacks that are applicable to transformer-based trackers have more impact on the object masks accuracy rather than on the accuracy of the bounding boxes on VOT2022ST dataset (Section 4.1).“
>
> And:
>
> “According to the accuracy scores of Table 2, the object binary mask was more affected by the adversarial attacks than the object bounding box. However, the difference of drop percentages in assessing the predicted bounding boxes and binary masks are negligible for the EAO and Robustness metrics of TransT-SEG tracker.“
>
> And:
>
> “The binary masks predicted by the MixFormerM tracker after the IoU attack for all computed metrics dropped greater than the metrics computed for the predicted bounding boxes. Nonetheless, following the CSA attack, the drop percentages of metrics for MixFormerM outputs were smaller and more consistent compared to those resulting from the IoU attack. “

---

> ### Author Response · Authors · 2024-04-21
> **Q3. Requested Changes**
>
> **“Requested Changes:
> Critical:
> I'd suggest authors add a comparison between models with different backbones (Transformer-based v.s. non-Transformer-based) with a detailed analysis.
> I'd also suggest authors analyze how Transformer makes a difference here.“**
>
> In response to the reviewer's feedback, we conducted an additional experiment to illustrate the adversarial robustness of both transformer and non-transformer trackers. Subsequently, we analyzed the findings of this evaluation. Our experiments reveal that although transformer trackers, containing ROMTrack, MixFormer and TransT, exhibit more robust performance before and after the adversarial attacks, comparing the percentage drops from original scores indicates that these trackers’ average overlap decreased from the original scores greater than almost all other trackers after the IoU attack.  For further elaboration, please refer to Section 4.4.

---

### Review · Reviewer_5JXt · 2024-03-14

**Summary Of Contributions:**

This work studies the reproducibility of four existing adversarial tracking attacks against three transformer trackers while existing works do not include these studies.  Specifically, the paper implements four attacking methods including SPARK, CSA, IoU, and RTAA against two transformer-based trackers and their variants including TransT & TransT-SEG, and MixFormer & MixFormerM. To study the effectiveness of attacks, the paper runs attacks and trackers on two small datasets, i.e., VOT2022 and UAV123. The results provide some observations of different attacks and demonstrate their effectiveness. The work considers three attacking scenarios including Adversarial Attacks under Tracker Output, Perturbation Level, and Upper Bound.

**Audience:**

Yes

**Broader Impact Concerns:**

Visual object tracking is still an important task in computer vision. Existing works have found that deep trackers are also vulnerable to adversarial attacks. Existing works mainly focus on Siamese-based trackers instead of the state-of-the-art transformer trackers. The work could fill the gap.

**Claims And Evidence:**

Yes

**Requested Changes:**

Please refer to the weakness of the "Strengths and Weakness" section.

**Strengths And Weaknesses:**

Strengths:
1. This work fills the gap between state-of-the-art trackers and adversarial tracking attacks. Existing attacks mainly work on SiamRPN-based trackers due to their easy implementations with different backbones.
2. The results provide some new observations on the existing attacking method.
3. The work could push the progress of adversarial tracking attacks by revealing some issues.

Weakness:
1. It does not indicate the main difference in attacking results on previous trackers and transformer trackers. The work only runs attacks against transformer-based trackers while ignoring other trackers. Readers cannot know the differences and challenges of attacking transformer trackers.
2. The work only runs attacks on two small tracking datasets, which can hardly provide solid and convincing conclusions.
3. The work did not introduce the attacks and trackers comprehensively. I suggest the authors detail trackers and unify the formulations of trackers and attacks, which can provide an intuitive understanding.
4. The experiment setups are not explained. The paper provides three experimental setups however why adopting these setups are not explained. Some tables only contain two attacks or one tracker instead of all attacks against all trackers. Why?
5. Transferability is a key issue in attacking. There are few discussions on this part. Please consider adding a section to discuss the transferability.

---

> ### Author Response · Authors · 2024-04-21
> **Q1. Transformer versus non-transformer performance after adversarial attacks**
>
> We would like to thank the reviewer for the useful comments. We are glad to mention that our work fills the gap between novel transformer trackers and the existing adversarial attacks for visual trackers. By presenting new observations on attack methods, we state new issues to push the progress on the field.
>
> **“It does not indicate the main difference in attacking results on previous trackers and transformer trackers. The work only runs attacks against transformer-based trackers while ignoring other trackers. Readers cannot know the differences and challenges of attacking transformer trackers.”**
>
> This revised version of the paper now includes a thorough comparison with non-transformer trackers including Siamese-based and discriminative trackers. Our analysis reveals that Although transformer trackers, containing ROMTrack, MixFormer and TransT, exhibit more robust performance before and after the adversarial attacks, comparing the percentage drops from original scores reveals that these trackers’ average overlap decreased from the original scores greater than almost all other trackers after the IoU attack (Section 4.4).
>
> In Sec. 4.4, we explained the experiment of transformer and non-transformer trackers compared before and after the adversarial attacks applied to them.

---

> ### Author Response · Authors · 2024-04-21
> **Q2. Tracking datasets**
>
> **“The work only runs attacks on two small tracking datasets, which can hardly provide solid and convincing conclusions.”**
>
> We used two datasets, VOT2022 and UAV123, for our previous version of the manuscript and added one new dataset, GOT10k dataset set, to the new revised version.
>
> We adopted  VOT2022 dataset for the following reasons:
> 1. Unlike other datasets that use the one-pass evaluation protocol, the VOT2022 follows the anchor-based short-term protocol for the trackers evaluation. The metrics of this protocol present another view of the trackers' performances. As it is stated in the manuscript, the effectiveness of SPARK attack in this situation is better than in the one-pass evaluation protocol used in experiment #2.
> 2. The VOT2022 provides the evaluation based on both object bounding boxes (STB) and binary masks (STS) which makes it a fair setup for our experiment #1.
> 3. Besides, the newest trackers are annually assessed by the VOT community and the most robust trackers are listed and announced on different sub-challenges such as short-term sub-challenge of VOT2022. From the announced list, we chose the most robust transformer trackers for our experiments such as MixFormerM, which was among the top five list of VOT2022 for binary mask prediction.
>
> We also selected the UAV123 dataset and GOT10k dataset for our second, third, and forth experiments. The UAV123 dataset includes 123 video sequences with axis aligned bounding box annotations and 112578 video frames. The GOT10k dataset contains 180 video sequences and 21007 video frames. We used this dataset to compare different trackers' performance against adversarial attacks which is fully explained in the new version of manuscript, Section 4.4.

---

> ### Author Response · Authors · 2024-04-21
> **Q3. Details  of object trackers and attack methods**
>
> **“The work did not introduce the attacks and trackers comprehensively. I suggest the authors detail trackers and unify the formulations of trackers and attacks, which can provide an intuitive understanding.”**
>
> Thanks to the reviewer's comment, we thoroughly revised the “Related works” and “Object Trackers and Adversarial Attacks” sections to provide a more detailed and unified explanation of the trackers and attacks.

---

> ### Author Response · Authors · 2024-04-21
> **Q4. Experiment setups and transferability**
>
> **“The experiment setups are not explained. The paper provides three experimental setups however why adopting these setups are not explained. Some tables only contain two attacks or one tracker instead of all attacks against all trackers. Why?”**
>
> It’s because not all attacks can be applied to all trackers. The attack methods use some features of tracker output as the attack proxy such as classification labels. Therefore, if the trackers do not infer or work on the attack proxy used to generate the adversarial perturbations, the attack algorithm is not applicable to those trackers. For instance, the RTAA attack employs classification and regression labels to compute the adversarial loss and create the adversarial frame patch. So, for trackers that do not contain the classification and regression labels in their output such as ROMTrack, DiMP or MixFormer, the RTAA attack is not applicable as a white-box attack. In our manuscript, we aimed to apply the adversarial attacks in their original setting as they are proposed in the original papers. A new subsection entitled “Transferability” is added to the manuscript, Sec. 3.3, to explain these setups and choices.
>
> **“Transferability is a key issue in attacking. There are few discussions on this part. Please consider adding a section to discuss the transferability.”**
>
> As mentioned above, a new subsection entitled “Transferability” is added to the manuscript, sec. 3.3, to explain these setups and choices, thanks to the reviewer's feedback.

---

### Review · Reviewer_7sp3 · 2024-03-23

**Summary Of Contributions:**

This paper presents a study on the reproducibility of existing attack approaches against transformer trackers, aiming to understand their behavior under adversarial conditions and how these attacks affect tracking accuracy. By conducting experiments across different perturbation levels on the VOT2022ST and UAV123 datasets, it explores the robustness of transformer trackers to adversarial perturbations, focusing on object-bound box and binary mask outputs. The study shows that transformer trackers with stronger cross-attention modeling show increased robustness against adversarial attacks, particularly noting that the impact of perturbations varies between object masks and bounding boxes. Furthermore, it shows that while some attack methods result in temporally sparse and imperceptible perturbations, others lead to more perceptible noise, significantly affecting tracking performance and indicating a need for new attack strategies to challenge advanced transformer trackers effectively.

**Audience:**

Yes

**Broader Impact Concerns:**

This paper presents a study on the reproducibility of existing attack approaches against transformer trackers, aiming to understand their behavior under adversarial conditions and how these attacks affect tracking accuracy. It does not raise broader impact concerns related to societal, ethical, or environmental issues.

**Claims And Evidence:**

Yes

**Requested Changes:**

### Major :
1. Attack Methods and Section 4.2:
The manuscript could benefit from a more detailed mathematical exposition in the section on attack methods specifically in Section 4.2.
2. Choice of Trackers for Evaluation:
The rationale behind selecting TransT-SEG and TransT for evaluation needs further clarification. Given that Chen et al., 2023, indicate TransT-SEG's superior performance over TransT, an explanation for not considering other potentially more competitive trackers would be insightful.
3. VOT2022 Baseline and Tracker Selection:
The justification for choosing VOT2022 as the baseline, particularly focusing on the short-term challenge in both the STB and STS cases, is not clear. An explanation of this choice would help understand the context and relevance of the evaluation criteria.
Additionally, elaborating on the selection of MixFormerL, a top-ranked tracker in the VOT-STS2022 challenge and STB challenge, over other trackers would provide clarity on its inclusion criteria.
4. Discussion on Attack Applicability:
The assertion that black-box attacks are the only viable methods against transformer trackers with deep relation modeling, and the subsequent claim regarding the inadequacy of existing attack methods, lacks convincing reasoning. More rigorous evidence or argumentation may be required to substantiate these claims.


### Minor :
Writing issues:
* While the introduction lists findings, this paper lacks a high-level summary of its contributions.
* The Related Work section is not divided into subsections. Organizing it into categories such as trackers, white-box methods, and black-box methods would enhance readability.
* Section 5 could benefit from subdivision into separate subsections. Currently, the result report, discussion of results, and conclusion are intertwined, making it challenging to discern the authors' valuable discussions.
* The authors should unify the use of ‘modelling’ and ‘modeling’

**Strengths And Weaknesses:**

### Strengths:
* The paper targets a critical issue in its field of study.
* The Investigation section presents experimental data in detail.

### Weaknesses:
* For the attack methods and Section 4.2, including more mathematical details would make it more convincing and facilitate our understanding of the background.
* There needs to be further justification for choosing TransT-SEG and TransT together. Given that Chen et al., 2023 demonstrated TransT-SEG's superiority over TransT, why were other trackers not considered?
* The choice of VOT2022 as the baseline requires more explanation. Why was it selected as the baseline? Additionally, the decision to focus on the short-term challenge in both STB and STS cases needs clarification. Also, the rationale behind selecting MixFormerL, a top-ranked tracker in the VOT-STS2022 challenge and STB challenge, over others, is not clear.
* In the discussion, the authors assert that "black-box attacks are the only applicable methods on transformer trackers with deep relation modeling" and claim "these trackers were not sufficiently challenged by existing attack methods." However, the reasons provided in the discussion do not convince me of the study's rigor.
* Writing issues need to be addressed.

---

> ### Author Response · Authors · 2024-04-21
> **Q1. Mathematical details of trackers and attack methods**
>
> We express our gratitude to the reviewer for precise and insightful comments. We are pleased to work on a critical issue in the field of visual tracking. Our goal is to contribute to the field by presenting our experimental results and pointing out the existing challenges for future works.
>
> **“Attack Methods and Section 4.2: The manuscript could benefit from a more detailed mathematical exposition in the section on attack methods specifically in Section 4.2.”**
>
> Thanks to the reviewer's comment, we refer the reviewer to check the new version of the manuscript in which we re-organized the related works, the attack and tracker descriptions and Section 4.2.

---

> > ### Author Response · Authors · 2024-04-21
> > **Q3. VOT2022ST dataset**
> >
> > **“VOT2022 Baseline and Tracker Selection: The justification for choosing VOT2022 as the baseline, particularly focusing on the short-term challenge in both the STB and STS cases, is not clear. An explanation of this choice would help understand the context and relevance of the evaluation criteria. Additionally, elaborating on the selection of MixFormerL, a top-ranked tracker in the VOT-STS2022 challenge and STB challenge, over other trackers would provide clarity on its inclusion criteria.”**
> >
> > We chose the VOT2022 Short-term dataset and protocol for the first experiment because of these three reasons.
> > 1. Unlike other datasets that use the one-pass evaluation protocol, the VOT2022 follows the Anchor-based Short term protocol for the trackers' evaluation. The metrics of VOT2022 baseline are obtained from the anchor-based short-term protocol which presents another view of the trackers' performances. As it is stated in the manuscript, the effectiveness of SPARK attack in this protocol is better than in the one-pass evaluation protocol which is used in experiment #2.
> > 2. The VOT2022 provides the evaluation based on both object bounding boxes (STB) and binary masks (STS) which makes it a fair setup for our experiment #1.
> > 3. Besides, the newest trackers are annually assessed by the VOT community and the most robust trackers are listed and announced on different sub-challenges such as short-term sub-challenge of VOT2022. From the announced list, we choose the most robust transformer trackers for our experiments such as MixFormerM which was among the top five list of VOT2022 for binary mask prediction.
> >
> > Additionally, we explained the reason for choosing the MixFormer tracker and its role on the VOt2022 ST sub-challenge in the manuscript. According to the constructive comments of the reviewer, the revised manuscript is updated to include this debate in Section 4.1, Evaluation Protocol part as follows:
> >
> > “We selected the VOT2022 Short-term dataset and protocol (Kristan et al., 2023) because of these three reasons. Unlike other datasets that use the one-pass evaluation protocol, the VOT2022- ST follows the anchor-based short term protocol for the trackers' evaluation. The metrics of VOT2022 baseline are obtained from the anchor-based short-term protocol which presents another view of the trackers’ performances. Also, the VOT2022-ST provides the evaluation based on both object bounding boxes (STB) and binary masks (STS) which makes it a fair setup for our experiment. Therefore, the experiments for different attacks are achievable offline. Besides, the newest trackers are annually assessed by the VOT community and the most robust trackers are listed and announced on different sub-challenges. For instance, the MixFormerM tracker (Cui et al., 2022) was among the five top-ranked trackers for binary mask prediction and its other variant, MixFormerL won third place on the bounding box prediction sub-challenge.”

---

> > ### Author Response · Authors · 2024-04-21
> > **Q4. Discussion on Attack Applicability**
> >
> > **“Discussion on Attack Applicability: The assertion that black-box attacks are the only viable methods against transformer trackers with deep relation modeling, and the subsequent claim regarding the inadequacy of existing attack methods, lacks convincing reasoning. More rigorous evidence or argumentation may be required to substantiate these claims.”**
> >
> > Thanks to the reviewer’s comment, we updated the manuscript with specific Sec. 3.3, entitled ‘Transferability’ to explain the concerns related to the attack applicability. Also, we revised the general claims with more specific sentences to clear our message and precise the claims.
> > In Transferability, Section 3.3:
> >
> > “In our study, we applied the attacks as they are proposed in the original works. For instance, the SPARK attack (Guo et al., 2020) was originally developed to attack the trackers via a white-box setting. Therefore, we apply this attack as a white-box attack in our experiments. As the SPARK algorithm uses both classification and regression labels to generate the perturbation, for some trackers such as (Cui et al., 2022; Cai et al., 2023; Bhat et al., 2019; Danelljan et al., 2020), it is not applicable in a white-box setting. In Table 1, we specified the applicable attacks on visual trackers in our study. For instance, in MixFormer (Cui et al., 2022), the classification labels are fused by a score prediction module to infer one single score as the part of output.“
> >
> > In Discussion, Section 5:
> >
> > “In our settings, the only applicable attacks against MixFormer and ROMTrack are black-box attacks, i.e. IoU and CSA. Among black-box methods, the IoU attack outperformed CSA for TransT, MixFormer, Mix-FormerM and ROMTrack trackers, Sections 4.1 and 4.4. However, the effect of the IoU method against ROMTrack and MixFormer’s are trivial, Section 4.4. The ROMTrack and MixFormer bounding box predictions were harmed by the IoU method up to 18.11 % and 18.81% on GOT10k and VOT2022 datasets for the average overlap metric, respectively. This indicates that these trackers were not being challenged enough with existing applicable attack methods..“

---

> ### Author Response · Authors · 2024-04-21
> **Q2. TransT versus TransT-SEG performance on different datasets**
>
> **“Choice of Trackers for Evaluation: The rationale behind selecting TransT-SEG and TransT for evaluation needs further clarification. Given that Chen et al., 2023, indicate TransT-SEG's superior performance over TransT, an explanation for not considering other potentially more competitive trackers would be insightful.”**
>
> We selected the TransT tracker in the second experiment where the goal is to study the impact of white-box attacks. The TransT tracker pioneered the use of transformer networks for tracking, so it is important to be included in our study. However, we investigated the difference between TransT and TransT-SEG performance on UAV123 dataset further in detail to reassure us about choosing the most comparative tracker.
>
> We assessed TransT-SEG without any attacks on the UAV123 dataset, our test set in the second experiment. As the scores are presented below, the TransT-SEG performance is below that of TransT on the UAV123 dataset which is annotated by bounding boxes.
>
> |       | AUC | Precision |
> | ----------- | ----------- | ----------- |
> | TransT      | 68.08      | 87.61      |
> | TransT-SEG   | 67.00        | 85.98       |
>
> Therefore, the best performance on the UAV123 dataset between TransT and TransT-SEG belongs to the TransT tracker.
>
> Furthermore, we checked the TransT-M paper (Chen et. al, 203) to find the results related to TransT-SEG superiority over TransT. In Table 10 of the TransT-M paper, the ablation study performed on VOT2021-Short term shows greater scores for TransT-SEG in comparison to the TransT tracker. Since VOT2021-ST has the same data and protocol as VOT2022-ST, our test in the first experiment, we could compare our results to Table 10 of the TransT-M paper.
>
> We believe the discrepancy between our results and those reported in (Chen et al., 2023) are due to the fact that they report scores on bounding box evaluation for TransT tracker and segmentation masks for TransT-SEG tracker. We obtained almost the same results in our first experiment for the TransT-SEG tracker. When the evaluation is based on the bounding boxes (STB), TransT-SEG scores (Table 1 of our manuscript, STB ‘clean’ scores) are similar to the TransT tracker (Table 10, TransT scores). On the other hand, TransT-SEG scores of our experiment (Table 1 of our manuscript, STS ‘clean’ scores) are the same scores as those reported for TransT-SEG in Table 10 of the TransT-M paper.
>
> In addition, Table 11 of TransT-M paper (Chen et al., 2023) reports the evaluation scores for TransT and TransT-M, and completely ignores the TransT-SEG performance on NFS, OTB100, and UAV123 datasets. According to our experiments, the performance of TransT-SEG on the UAV123 dataset, which is our test set on experiment 2, is below that of the TransT tracker.
>
> Moreover, we found TransT performance better than TransT-SEG in predicting bounding boxes on the GOT10k test set, our fourth experiment. The obtained results without applying any attacks on GOT10k are as follows:
>
>
> |       | AO | SR$_{0.5}$ | SR$_{0.75}$ |
> | ----------- | ----------- | ----------- |----------- |
> | TransT      | 0.723     | 0.823      | 0.682      |
> | TransT-SEG   | 0.719     | 0.816      | 0.680      |
>
>
> Therefore, we conducted the last experiment with the TransT tracker.

---

### Author Response · Authors · 2024-05-24
**Camera Ready Version**

Dear Action Editor and reviewers,

We would like to sincerely thank you for your constructive feedback throughout the rebuttal process. Following your suggestions, we updated the manuscript to address the first two points in the following way.

+ The name of Sec. 3.3 is changed to “Attack Setups”. We agree that this section is not much about transferability and that experiments on transferability are beyond the scope of our paper.

+ The first paragraph of Sec. 3.3 in the previous version is now the third paragraph at the end of Sec. 2.2. This brings up this important explanation of attacks on vision transformers earlier in the paper.

+ Since the tracker and attack methods are fully explained in Sec. 3, we considered Sec. 3.3 as a description of our attack setups. We fully revised the text of Sec. 3.3 accordingly, and to clarify the explanations on white-box vs black-box attacks, and reduce the emphasis on transferability, which is out of scope for the experiments in our paper.

+ Specifically, regarding the second point, we change the aforementioned sentence to (bold part added):

“Consequently, white-box attacks utilizing these intermediate outputs (namely, object candidates and their labels) to compute the adversarial loss are no longer applicable to the new transformer backbones **via the transformer's gradients themselves. Although they can be transferred in a black-box way (i.e., making adversarial samples with other backbones or other losses), our focus is to employ the transformer gradients in generating adversarial examples in a white-box setting.”**


We have submitted the camera-ready version containing the revised manuscript, supplementary material and the link to the codes.

---

> ### Comment · Action_Editor_gjQz · 2024-05-25
> **Minor Suggestion**
>
> Dear authors,
>
> A very minor point, but on p17 you still write "Transferrability section 3.3" which should be edited due to the section name change.  Also replace "section" by "Section".
>
> Also, at the end of the abstract, instead of just a hyperlink, I suggest displaying the URL so that it can be read even by those that print the paper.  (But the URL can still be clickable for online readers.)
>
> Action Editor

---

> > ### Author Response · Authors · 2024-05-25
> > **Manuscript is updated**
> >
> > Dear Action Editor,
> >
> > We would like to thank you for your suggestions. We have updated the manuscript according to your comments.

---

### Decision · Action_Editor_gjQz · 2024-05-21

**Recommendation:** Accept with minor revision

**Comment:**

The reviewers were generally satisfied with the changes made following the reviews and response period.  However, one reviewer noted two points:
- Concern that the "transferability" section may need major editing.  They suggested that discussion on attack setups would be more suited to earlier parts of the paper, and noted a lack of discussion on transferring attacks (i.e., attacking one model and then transferring that attack to another model).  Thus, they questioned why the subsection is titled "Transferability".
- They claimed that the following sentence is incorrect: "Consequently, white-box attacks utilizing these intermediate outputs (namely, object candidates and their labels) to compute the adversarial loss are no longer applicable to the new transformer backbones.", adding that as long as adversarial examples can be formed, transferability experiments should be possible.
- They noted recent new defense methods such as "LRR: Language-driven Resamplable Continuous Representation Against Adversarial Tracking Attacks"

The third point above can be disregarded (or mentioned briefly if the authors wish), as it is not appropriate to insist on being up-to-date with work that was released during the reviewing period.  But I would like to ask that the authors make careful revisions to address the first two points.  Experiments on tranferrability are not mandatory (e.g., you can mention that they are beyond your scope), but all claims within Section 3.3 should be correct and appropriately placed.

**Audience:**

The broad themes such as adversarial attacks and transformer-based methods are major topics in the machine learning community.  While trackers are a narrower topic, it is still of interest to many researchers, and it is likely that some of them will appreciate a reproducibility study of this kind.

**Claims And Evidence:**

This is a reproducibility paper, seeking to study the reproducibility of various attacks against transformer trackers and the resultant impact on tracking accuracy.  Diverse empirical studies are performed on various trackers and data sets.  The attacks are compared along a number of criteria and their advantages/disadvantages are discussed, as summarized in Section 5.